



# Identification, Mapping and Eco-hydrological Signal Analysis for Groundwater-dependent Ecosystems (GDEs) in Langxi River Basin, North China

Mingyang Li, Fulin Li, Shidong Fu, Huawei Chen, Kairan Wang, Xuequn Chen, Jiwen Huang

Water Resources Research Institute of Shandong Province, Shandong Provincial Key Laboratory of Water Resources and Environment, Jinan 250013, China

*Correspondence to*: Fulin Li (fulinli@126.com)

**Abstract.** Groundwater-dependent ecosystems (GDEs) refer to ecosystems that require access partially or completely to groundwater to maintain their ecological structure and functions, provisioning very important services for the health of land, water, and coastal ecosystems. However, regional identification of GDEs is still difficult in areas affected by climate change

and extensive groundwater extraction. To address this issue, taking the Langxi River Basin (LRB), one of the lower tributaries of the Yellow River in North China, as an example, we propose a four-diagnostic criteria framework for identifying the GDEs based on remote sensing, GIS data dredging and hydrogeological surveys. Firstly, the potential GDEs distributions are preliminarily delineated by the topographic features and the differences of vegetation water situation, soil

moisture in the end of the dry and wet seasons. On this basis, according to the given GDEs identification criteria, three main types of GDEs in the basin including the stream-type GDEs (S-GDEs), vegetation-type GDEs (V-GDEs) and karst aquifer-type GDEs (K-GDEs) are further determined by comparing the relationship between groundwater table and riverbed elevation, vegetation root development depth, and though surveys of karst springs and aquifers. And then the GDEs are mapped using the spatial kernel density function which can represent the characteristics of spatial aggregation distribution.

Results show that the potential GDEs are mainly distributed in plain areas and a small part in hilly areas, reflecting the moisture distribution status of waters, vegetation and wetlands in the basin that possibly receive groundwater recharge, however, the true GDEs are concentrated in the riverine and riparian zone, the vegetation-related wetland and the scattered karst spring surroundings which groundwater directly moves toward and into. To verify the reliability of GDEs distributions, ecohydrological signal analysis were also performed in this paper. The analysis of river hydrological process curve and karst

spring hydrograph in Shuyuan section showed that the proportion of base flow to river flow is about 54.15% and S-GDEs still receive spring water recharge even in the extremely dry season. And the analysis of hydrochemical sampling from the karst aquifer, Quaternary aquifer, spring water and surface reservoir water reveals that GDEs are also relished by groundwater. More important, we also found a distinctive ecohydrological signal of GDEs is the presence of millimeter-sized groundwater fauna living in the different types of GDEs. Finally, the validity of the method proposed in the study for

identification and mapping of the GDEs is also discussed. It still has some room for improvement if the water, sediments and biotic connectivities between groundwater and GDEs are analyzed by using isotopes and DNA technology under the recommended four-diagnostic criteria framework.





**Keywords:** Groundwater-dependent Ecosystems (GDEs); Identification and mapping; Hydrograph separation; Hydrochemical clustering; Groundwater faunas; Eco-hydrological Signals

**1 Introduction**

In the area where surface water interacts with groundwater, due to the temporal and spatial differences in precipitation, infiltration, recharge, runoff and other processes, an ecosystem is formed around low-lying land, riverbanks on both banks and karst caves. This is due to temporal and spatial differences in precipitation, infiltration, recharge, runoff, and other processes (Bowles and Arsuffi, 1993; Rohde et al., 2017). Early scholars called this ecosystem as "Groundwater-fed
wetland" or "Groundwater-dominated stream ecosystems" (Eamus and Froend, 2006; Gilvear et al., 1993; Petts et al., 1999). Australian scholars proposed the concept of "groundwater-supported ecosystems" or "groundwater-dependent ecosystems" (GDEs for short) earlier, with a focus on the water requirements of plants (Clifton and Evans, 2001; Hatton et al., 1997). These ecosystems have distinct characteristics that are closely related to groundwater on a continuous basis and may also be seasonally or occasionally dependent on it (Foster et al., 2010). The composition, structure, and function of GDEs are
influenced by groundwater through flow, nutrient recharge, and pressure and water temperature transport. Additionally, the biological processes of GDEs, such as plant photosynthesis, microbial action, and animal activities, can impact surface water-groundwater hydrological processes, including surface water evaporation, flow, groundwater seepage, and recharge (Boulton et al., 2003; Murray et al., 2003; Schenková et al., 2018). GDEs not only sustain the health of ecosystems but also mitigate the effects of floods and droughts while providing essential products and services, such as food production and
water purification, to humans. Consequently, enhancing their protection and management is of vital ecological, social, and economic significance.

Scholars have classified GDEs in numerous ways over the years. In 1997, Hatton et al. (1997) proposed a classification system for Australia's GDEs based on their reliance on groundwater, dividing them into four categories: wetlands and terrestrial GDEs, mound springs ecology GDEs, aquatic GDEs, and aquifer and cave GDEs. Building on Hatton et al. (1997)
classification system, Clifton and Evans (2001) introduced two additional types of GDEs: terrestrial fauna GDEs and estuarine and near-shore marine GDEs, based on the spatial distribution of GDEs in land and estuary areas. In later studies, Eamus and Froend (2006) simplified the six classifications into three (groundwater ecosystems, ecosystems dependent on belowground expression of groundwater, and ecosystems dependent on aboveground expression of groundwater), while Bertrand et al. (2012) argued that as a result of climate change, it is necessary for GDEs to define potential GDE ecosystems
that can account for the full spectrum of GDEs under varying climatic conditions. The Global Water MATE Core Group, World Bank, and United States Department of Agriculture's (USDA) National Agricultural Statistics Service (NASS) guidelines have classified GDEs based on differences in arid, humid, coastal, and inland areas (Foster et al., 2006). As evident from the aforementioned guidelines, the classification of GDEs varies across different research regions and





environments. Therefore, defining the appropriate classification of GDEs is crucial in accurately identifying and distributing
them in the study area.

GDEs are vulnerable to disturbances caused by both natural and human activities (Dong and Zhang, 2011; van
Engelenburg et al., 2018). Natural factors that affect GDEs include climate change, topography, and hydrogeological
conditions. Human activities that affect GDEs include changes in underlying surface conditions, habitat environment
disturbances, water conservancy construction, and groundwater exploitation. The impact of climate change and groundwater
exploitation on GDEs has garnered increasing attention, and some scholars have conducted multi-factor impact assessments
and risk analyses to better understand their effects. According to Humphreys (2006) , aquatic vegetation is closely linked
with groundwater ecosystems. Laio et al. (2009) developed a conceptual model to illustrate the relationship among rainfall,
groundwater level, and vegetation in the GDEs. Their model indicated that the stochastic, dynamic changes in groundwater
level are closely tied to climate change, vegetation coverage, and water resource management levels. Hancock and Boulton
(2009) conducted multidisciplinary research on aquifers, hydrogeology, ecology, and the relationship between groundwater
and its associated ecosystems. They noted that surface vegetation is also influenced by groundwater processes, which are
crucial to consider when studying GDEs, particularly in water-limited environments.

Identifying and mapping GDEs in the wild can be challenging, particularly in areas heavily impacted by human
activities, despite their widespread distribution. Early studies primarily utilized field hydrogeological and ecological survey
techniques to determine GDEs. For instance, Eamus and Froend (2006) suggested a toolbox comprising various methods to
identify GDEs and their reliance on groundwater, as well as vegetation processes dependent on groundwater, based on the
type of GDE. With the widespread adoption and application of remote sensing and geographic information system (GIS)
methods, researchers have been able to analyze spatial data with greater precision. For example, Howard and Merrifield
(2010) utilized GIS methods to study California, USA, and establish the Groundwater Dependency Index. By doing so, they
were able to identify, map, and classify various types of groundwater-dependent ecosystems (GDEs) such as springs,
wetlands, and streams into different dependency levels. Hoogland et al. (2010) evaluated the dry water shortage of GDEs in
the Netherlands by creating groundwater depth maps. Lastly, Gou et al. (2015) were the first to use GIS database information
to determine the potential distribution of GDEs at a state/province level. To track and identify changes in vegetation pixels,
researchers often use Landsat imagery to analyze the Normalized Difference Vegetation Index (NDVI), which helps
determine the distribution of groundwater-supported vegetation at the aquifer or watershed scale. While traditional
hydrogeological surveys can be time-consuming and expensive, remote sensing methods provide an efficient way to
determine large-scale GDEs distribution. However, remote sensing may not always be accurate at small scales, such as the
segment scale of river sections. Therefore, combining traditional hydrogeological surveys, field ecological monitoring,
global positioning systems (GPS), GIS, and remote sensing (RS) is an effective method for identifying and mapping GDEs.

To improve the identification and mapping of GDEs, it is important to analyze their ecohydrological characteristics.
This involves studying the interaction process between surface water and groundwater, as well as the simulation of material
transport to determine the regional ecohydrological characteristics. These characteristics can often be regarded as a specific



signal for monitoring ecosystem status and linking the functions of organisms to ecohydrological processes, such as the rhythm of hydrometeorological elements, hydrogeochemical characteristics, and biological indicators (biodiversity,
connectivity, etc.), etc. Understanding these characteristics can help determine if there is hydrological connectivity between groundwater and potential GDEs and whether a "hydrological continuum" can be formed. This is crucial for maintaining the integrity of the system and the ecological processes regulated by the "Ecotones". Hao et al. (2018) found that although groundwater development has weakened the relation between spring discharge and precipitation, the resonant frequency between spring discharge and precipitation remained unchanged by studying the discharge data of Niangziguan Spring, a
karst hydrological case in North China. Brancelj et al. (2020) provided an overview of groundwater fauna in the phreatic zone of the Classical Karst aquifer, and discovered the rate of endemism within the area is very high (around 50%), which can be considered as descriptors of aquifer type and habitat structure, as well as water flow regime and groundwater flow paths. These case studies are examples of unique ecohydrological habitats that are an essential part of global research. In addition, the ecohydrological process of GDEs is also demonstrated through the hydrological and hydraulic connections
between the vegetation ecosystem and precipitation, surface water, and groundwater. Therefore, analyzing the ecohydrological characteristics of GDEs can provide valuable insights into their functioning and can help in their effective management and conservation. Moreover, what sets GDEs apart from other ecosystems is the unique presence of groundwater invertebrate fauna, spanning from millimeters to centimeters, which serve as crucial indicator species for groundwater-supported ecosystems. Therefore, conducting sampling, species identification, assessing biodiversity and
ecology of groundwater faunas, and zoning animal habitats are all essential components of GDE researches. From the above analysis, it can be seen that the research on the distribution of GDEs is still in the initial stage of exploration, and the research methods are not the same. It is urgent to put forward a comprehensive and applicable research theory.

The author provided a comprehensive summary of research on GDEs (Li et al., 2018), revealing that there is limited research conducted in China, with most studies focusing on the hyporheic zone, karst ecotone, northwest grassland, desert
oasis, and other regions. Furthermore, the overexploitation of groundwater in northern China has resulted in the shrinking of GDEs over a large area, making identification and mapping challenging. The Langxi River Basin (LRB) is situated on the south bank of the Yellow River, on the north side of Mount Tai, and is part of a vast carbonate distribution area. The region is also located in the western part of Jinan city, where many springs have developed, giving rise to various types of GDEs, which are typical of northern China. Thus, the purpose of this study is to identify the types of GDEs affected by human
activities and delineate their scope to improve the basis for regional water resources planning and karst spring protection. The study has three primary objectives: first, to propose a criteria framework for identifying, mapping, and verifying GDEs; second, to identify and map the distribution of GDEs in a typical study basin using the soil moisture-based remote sensing method and the spatial kernel density function; and third, to verify the reliability of GDEs zoning through ecohydrological signal analysis in the river basin.





## 2 Materials and Methods

### 2.1 Study area

The Langxi River basin (LRB) is a typical karst basin situated in the southwest of Jinan, China, spans an area of 137.8 km$^2$ with a river length of 26.68 km. It is one of the lower tributaries of the Yellow River (Figure 1). LRB is characterized by a continental monsoon climate with an average annual rainfall of 604 mm, which mainly falls during the summer season. Precipitation from June to August contributes to 65% of the total annual rainfall. Consequently, the runoff of the Langxi River is highly unstable, with maximum flow reaching 159 m$^3$ s$^{-1}$ during the flood season, and a risk of no-flow during the dry season. In order to optimize water usage, small reservoirs and dams have been constructed in the upper and middle reaches of the river for irrigation purposes. The amount of surface water resources, groundwater resources and exploitable groundwater in the LRB are 11.5 million m$^3$, 24.82 million m$^3$, and 20.75 million m$^3$, respectively. There are two aquifers in the study area, one is Quaternary pore water aquifer, which is regarded as unconfined aquifer. The second is the Cambrian karst aquifer, which is regarded as a confined aquifer.

The southern region of LRB is characterized by higher elevation and is enclosed by mountains. The valley is positioned in the central area and predominantly comprises low hills and plains, with an average elevation ranging from 100 to 250 m. The lowest point within the basin is located at the confluence of the Langxi River and the Yellow River, with an altitude of 36 m. The valley is home to diverse vegetation such as swamp, meadow, riparian sparse forest, and shrubs, creating distinct habitat landscapes along the river course.

The Cambrian and Quaternary strata are widely distributed throughout the basin, with the surface lithology consisting of hard limestone, Cambrian Zhangxia Group, Quaternary alluvial accumulation layer, and river-lake facies sandy clay and gravel. The water-bearing rock group comprises the Quaternary loose porous rock aquifer and the Cambrian carbonate fissure karst aquifer. Due to differences in topography and geology, various rising and descending springs are formed across different regions. The piedmont fault zones and areas with thin Quaternary sediments are rich in karst springs, resulting in wetlands of different sizes and scenic landscapes. According to historical records, there are 34 springs in the basin. Riparian zones and wetlands with shallow groundwater support GDEs through groundwater seepage and karst springs. Based on GIS data and survey results, a hydrogeological profile was constructed perpendicular to the Langxi River and Shuyuan Spring (Figure 2). The Shuyuan Spring, located at the junction of the Quaternary and Cambrian strata, is formed in the Cambrian Oolitic limestone of the Zhangxia Group, which has well-developed karst fissures in the east. The groundwater head is approximately 20 meters higher than the surface, resulting in an artesian descending spring.

### 2.2 Framework for identification, mapping and verifying of GDEs

This paper identifies GDEs in LRB based on the aforementioned GDE classification and the actual situations. Four criteria are used to identify GDEs:



1) Karst springs and associated wetlands: These ecosystems include karst springs, groundwater seeps, sinkholes, karst aquifers, and wetlands formed around karst springs or fed directly by karst groundwater. Collectively, they are referred to as karst groundwater-dependent ecosystems (K-GDEs).

2) Gaining streams: These are streams or parts of streams where flow is solely or partly contributed by inflow of groundwater. Typically, the groundwater table is at or above the stream level and moves toward and into the river, forming a stream-related ecosystem.

3) River riparian zone: This zone is adjacent to gaining streams and is characterized by distinctive plant and animal communities that are directly or indirectly fed by groundwater. The above two of GDEs are mainly distributed in the river and on both sides, and are referred to as stream-type GDEs (S-GDEs).

4) Vegetation-related ecosystems: These ecosystems are home to vegetation that grows in areas with shallow groundwater tables that roots may access to store water or where the vegetation types are considered phreatophyte species. This type of ecosystem typically maintains greenery even during extreme dry periods, and is referred to as vegetation-type GDEs (V-GDEs).

Using these four identified criteria, we propose a diagnostic framework for data collection, identification, mapping, and verification of GDEs in LRB (Figure 3).

## 2.3 Identification and mapping of GDEs

### 2.3.1 Potential GDEs quantification

First, using digital elevation model (DEM) and slope (calculated by DEM), we can distinguish the plains and hills of the basin (Eq.1), and further divide the plains of shallow fissure rocks according to the surface lithology, which is the area with the conditions for the formation of GDEs.

$$grid_{plain} = grid_{(slope \leq \Delta_{slope}) \cap (elevation \leq \overline{elevation})} , \qquad (1)$$

where, $grid_{plain}$ represents the grid divided into plains; $\Delta_{slope}$ is the threshold of the maximum plain slope, in this paper we take $\Delta_{slope} = 10°$. The determination of this parameter can be manually adjusted based on one-third of the average slope of the watershed until the plains and mountains are clearly distinguished. $\overline{elevation}$ is the average elevation of the basin. When applying this method in a basin with a significant difference in elevation, we highly recommend adjusting the value instead of simply using the average without consideration.

Previous studies utilized Eq. 2 and Eq. 3 to calculate the normalized difference vegetation index (NDVI) and the normalized water body index (NDWI) from infrared optical remote sensing Landsat satellite data, respectively. These indices were then used at the end of the wet and dry seasons to distinguish the rate of vegetation loss due to water and identify the extent of GDEs. Due to the difficulty of obtaining a clear NDWI image in the study area (Supplementary Figure 1), we opted to enhance our method by utilizing the more discriminative difference between the WET index and the





normalized difference built-up and soil index (NDBSI). The WET and NDBSI indices represent regional humidity and dryness, respectively. The two indices are the average values derived from multiple images captured during both the dry and wet seasons within a year. Additionally, the two largest sources of water supply, apart from precipitation, are the lateral input

of groundwater and rivers, which are dependent on the surface lithology of the basin. The difference between the two indices can be used to calculate the extreme variation in dryness and humidity in a specific region over a given period, thereby reflecting the average water reserves in that quarter. This is particularly relevant in areas such as river riparian zones, which are located near gaining streams and are characterized by distinct vegetation that remains green even during extremely dry periods. The potential GDEs distribution area can be determined by establishing a critical threshold ($\delta_{WD}$) of the difference.

The equations (Eq.2 to Eq.6) below illustrate the indices and methodology for assessing the distribution area of potential GDEs (Gao, 1996; Karbalaei Saleh et al., 2021; Pettorelli et al., 2005):

$$NDVI = \frac{\rho_{nir}+\rho_{red}}{\rho_{nir}-\rho_{red}}, \tag{2}$$

$$NDWI = \frac{\rho_{nir}+\rho_{swir1}}{\rho_{nir}-\rho_{swir1}}, \tag{3}$$

$$WET = c_1 \cdot \rho_{blue} + c_2 \cdot \rho_{green} + c_3 \cdot \rho_{red} + c_4 \cdot \rho_{nir} + c_5 \cdot \rho_{swir1} + c_6 \cdot \rho_{swir2}, \tag{4}$$

$$NDBSI = \frac{IBI+SI}{2} = \frac{\frac{\frac{2\rho_{swir1}}{\rho_{swir1}+\rho_{nir}}-\left[\frac{\rho_{nir}}{\rho_{nir}+\rho_{red}}+\frac{\rho_{green}}{\rho_{green}+\rho_{swir1}}\right]}{\frac{2\rho_{swir1}}{\rho_{swir1}+\rho_{nir}}+\left[\frac{\rho_{nir}}{\rho_{nir}+\rho_{red}}+\frac{\rho_{green}}{\rho_{green}+\rho_{swir1}}\right]}+\frac{[(\rho_{swir1}+\rho_{red})-(\rho_{nir}+\rho_{blue})]}{[(\rho_{swir1}+\rho_{red})+(\rho_{nir}+\rho_{blue})]}}{2}, \tag{5}$$

$$grid_{potenial\ GDEs\ area} = grid_{((WET-NDBSI)\leq\delta_{WD})}, \tag{6}$$

where, $c_1$ to $c_6$ are sensor parameters. Due to the different types of sensors, the parameters are also different. See Supplementary Table 1 for specific parameters. IBI and SI are building index and soil index. $\rho_{red}, \rho_{green}, \rho_{blue}, \rho_{nir}, \rho_{swir1},$ and $\rho_{swir2}$ are the red band, green band, blue band, near red band, mid infrared band 1, mid infrared band 2, respectively.

The $\delta_{WD}$ is the threshold used to demarcate the boundaries of potential GDEs distribution regions, which is determined by a trial-and-error algorithm. In this paper, we take $\delta_{WD} = 0.3$.

Modified by the GDE Mapping (GEM) method proposed by Barron et al. (2014), the potential GDEs based on vegetation and moisture responses observed at the surface during an extended dry period. It assumed that soil moisture would be depleted due to minimal rainfall over a six to seven-month drought, and areas with consistent greenness and high

surface wetness were more likely to have access to groundwater. Based on remote sensing indices, various regions can be categorized into different groups, such as permanent open water bodies, slow drying vegetation, fast drying vegetation, and crops. The permanent open water bodies exhibit consistently high wetness and consistently low greenness throughout the dry season. In contrast, the slow drying vegetation tend to display some level of reduction in both greenness and wetness after an extended dry period, typically caused by a decrease in groundwater contribution to base flow or annual groundwater table

subsidence. The fast-drying vegetation, where the root zone is frequently disconnected from groundwater, may experience a





significant decline in surface greenness and wetness due to the complete exhaustion of soil moisture stores at the end of a long dry period. Lastly, crop areas can be distinguished by discernible changes in planting and harvesting seasons.

By using riverbed and Quaternary water level data, it is possible to identify gaining streams, where the Quaternary water level is higher than the elevation of the riverbed, allowing groundwater to flow laterally and supply vegetation in the subsurface with water either seasonally or year-round. The areas where vegetation directly receives groundwater recharge can be determined by taking into account the vegetation root depth and Quaternary water level. These areas are referred to as V-GDEs. The riparian buffer zone of the river, which is nourished by groundwater, partially overlaps with these areas and is classified as S-GDEs.

By conducting field investigations and analyzing samples from karst springs and the surrounding environment, it is possible to define the distribution area of K-GDEs in the basin. Additionally, unique water chemistry and aquatic biological characteristics obtained through sampling, as well as changes in the surface vegetation and ecological environment recorded during the survey, can provide supporting evidence for the delineation of K-GDEs.

### 2.3.2 GDEs mapping

By applying the four diagnostic criteria, we are able to pinpoint the grids or ranges that are classified as GDEs. In this paper, we utilize the spatial kernel density algorithm to conduct a probability density partition of the distribution range of GDEs within the basin. The fundamental concept behind kernel density estimation is that each data point within the space will have an impact on a particular area through the density function. By constructing a spatial kernel density function model, we can determine the impact of any location within a given sample space (Cai et al., 2013; Hallin et al., 2004). This allows us to estimate the probability distribution of GDEs in our study. Kernel density estimation is a non-parametric technique used to estimate a probability density function. The kernel density at the coordinates $(x, y)$ can be calculated by Eq.7.

$$Density_{(x,y)} = \frac{1}{(radius)^2} \sum_{i=1}^{n} \left[ \frac{3}{\pi} \cdot pop_i \left( 1 - \left( \frac{dist_i}{radius} \right)^2 \right)^2 \right], dist_i < radius , \qquad (7)$$

where, $i = 1, 2, \ldots, n$ is the input point. Only include points in the sum if they are within a radius distance of the $(x, y)$ location. $pop_i$ is the population field value for point $i$, which is an optional parameter, in this paper, we define $pop_i = 1$. $dist_i$ is the distance between point $i$ and the $(x, y)$ position. $radius$ is the search radius in Eq.8, also known as the bandwidth. In this paper, the calculation method of bandwidth for the two adjacent grids is defined by taking the smaller value between the unweighted standard distances (Eq.9) and the second term of the $min$ function in Eq.8.

$$SearchRadius = 0.9 \times min \left( SD, \sqrt{\frac{1}{ln(2)}} \times D_m \right) \times n^{-0.2} , \qquad (8)$$

$$SD = \sqrt{\frac{\sum_{i=1}^{n}(x_i - \bar{X})^2}{n} + \frac{\sum_{i=1}^{n}(y_i - \bar{Y})^2}{n} + \frac{\sum_{i=1}^{n}(z_i - Z)^2}{n}} , \qquad (9)$$



where, $D_m$ is the (weighted) median distance from the (weighted) mean center. $SD$ is the standard distance.

By selecting a non-fixed bandwidth that varies based on the estimated location (balloon estimator) or sample points (pointwise estimator), we can utilize a powerful approach known as adaptive or variable bandwidth kernel density estimation. This method enables us to provide a more accurate depiction of the spatial distribution characteristics of GDEs.

## 2.4 Verifying GDEs using eco-hydrological signal analysis

By extracting and refining the eco-hydrological features of the basin, the eco-hydrological signal can be obtained and the discriminant range of GDEs can be verified. This paper selects three types of eco-hydrological signals: base flow recharge of groundwater to karst spring, hydrochemical characteristics of various water bodies in the basin, and groundwater fauna.

### 2.4.1 Hydrographic analysis: base flow signals

The base flow recharge of groundwater to karst spring can demonstrate that the karst aquifer in the watershed can replenish the surrounding ecosystem through surface runoff formed by spring water. The primary objective of the hydrological analysis method is to analyze and validate the proportion of groundwater recharge through base flow signals in S-GDEs. Based on the geometric characteristics of the runoff process line and hydrogeological expertise, the complete wave peak is segmented, and subsequently, the base flow is computed. The study utilizes the straight-line secant method, which involves horizontally dividing the peak of the flow process line using a horizontal line. It is stipulated that the contribution of surface runoff lies above the horizontal cutting line, while the contribution of base flow lies below the horizontal cutting line. The value of the horizontal line, which represents the runoff, can be determined as the minimum flow during the dry season, the minimum daily average flow during the dry season, or the minimum monthly average flow for the year. During the non-rainy season when the karst aquifer is recharged, the flow of the spring will gradually decrease in size until it matches the recharge rate of the aquifer, as there is no additional recharge from precipitation. The equation for the flow attenuation process can be written as (Rodríguez et al., 2017):

$$Q_t = Q_1 e^{-\alpha_1 t} + Q_2 e^{-\alpha_2 t} + \cdots + Q_n e^{-\alpha_n t} , \tag{10}$$

where, $Q_t$ is the total flow at time $t$, $Q_1$, $Q_2$, …, $Q_n$ are the 1 to $n$ decomposed replenishment items respectively; $\alpha_1$, $\alpha_2$, …, $\alpha_n$ are the parameters of the exponential regression model.

To gain a more comprehensive understanding of the eco-hydrological characteristics of karst-type GDEs, the study utilized Shuyuan Spring as a case study. Monthly average precipitation and spring flow data spanning 56 years (1990-2015) were collected, and typical years (July 1993 to July 1994) were selected using frequency ranking. The reasons for choosing the aforementioned time periods mainly stem from two factors. Firstly, these time periods fall within a natural state without any impact from groundwater exploitation. Secondly, there has been an extended duration of no rain recharge this year,





providing a unique opportunity to study the process of groundwater base flow recharge, which may not be available in other
years.

### 2.4.2 Hydrochemical analysis: water quality signals

The hydrochemical characteristics can distinguish whether a water body in the basin is recharged by the karst aquifer. The study collected water samples from 10 collection sites of three distinct water types: karst groundwater, Quaternary pore water, and surface water. Subsequently, all the water samples were passed through a 0.45 μm filter, and the liquid samples
were acidified to pH 2 using pure $HNO_3$ to prevent the precipitation of metals before metal analysis. The determination of basic metals was carried out using inductively coupled plasma mass spectrometry (ICP-MS, Agilent 7500C), while dissolved anions were analyzed using ion chromatography (IC, Metrohm 861). The primary ions and pollutants in the water were analyzed to determine their composition and content, which aids in understanding the water environment's condition in the hyporheic zone. To ensure test accuracy, three water samples were collected at each site for replication. The measured water
chemistry results were represented using a three-line diagram and clustered via Q-mode cluster analysis.

Q-mode cluster analysis is a principal component analysis method that focuses on variables. It uses distance measurements to determine the similarity between water quality indicators of various water samples and classifies them based on the distance between samples. In this research, we utilized the Euclidean distance and the shortest distance method for calculation.

### 2.4.3 Groundwater fauna sample analysis: stygofauna specie signals

One of the notable signs of GDEs is the presence of millimeter-scale groundwater fauna, which serves as a biological indicator of the groundwater ecosystem and helps confirm the identification and mapping of GDEs. These fauna also aid in determining the distribution of GDEs. In this study, we utilized stygofauna species signals from karst groundwater, surface water, and Quaternary pore water to verify different types of GDEs.

The general sampling methods are generally pumping sampling, net sampling and "unbaited traps" sampling (Hahn, 2005). In order to investigate the species and distribution of inverbrates in the ecosystem supported by karst groundwater, we designed a sampling method combining phreatobiological net sampling (Figure 4a), micro-pump sampling (Figure 4b) and unbaited trap sampler (Figure 4c). Before laboratory analysis, the samples were filtered through a 40-micron mesh, preserved in 5% formalin or 70% alcohol, and stained with rose bengal.

Benthic invertebrates were kept in 70% ethanol or 5% formalin, Zooplankton (Cladocerans, Copepods) were saved in 100 mL water sample with 4 to 5 mL formalin, Zooplankton (Protozoa, Rotifers) were saved in water samples with 1% (v/v) Lugol's solution, and fishes were stored with 10% formalin. The morphological characteristics of the stygofaunas were observed by stereoscopic biological microscope (Olympus SZ61).





## 2.5 Data

In this paper, the data mainly includes remote sensing data, and hydrogeological survey data (see Supplementary Table 2 for data sources). The remote sensing data used is the Landsat series of satellite datasets, Landsat Collection 2, the second major reprocessing effort on the Landsat archive, resulted in several data product improvements that applied advancements in data processing and algorithm development. These images contain 5 visible and near-infrared (VNIR) bands and 2 short-wave infrared (SWIR) bands processed to orthorectified surface reflectance, and one thermal infrared (TIR) band processed

to orthorectified surface temperature (Cook et al., 2014). We use Google Earth Engine (GEE) to handle and calculate remote sensing data, mainly including image merging, cropping, and cloud removal, et al. According to the precipitation in the study area, we selected the dry season and rainy season from December 2020 to March 2021 and from April to October 2021, respectively. Correspondingly, the time of our field investigation is consistent with the time of remote sensing imaging.

The elevation data used is the Shuttle Radar Topography Mission (SRTM, see Farr et al. (2007)) digital elevation data,

an international research effort that obtained digital elevation models on a near-global scale, and the grid slope is calculated from the digital elevation. This SRTM V3 product (SRTM Plus) is provided by National Aeronautics and Space Administration Jet Propulsion Laboratory (NASA JPL) at a resolution of 1 arc-second (approximately 30m).

Field survey mainly includes hydrogeological survey and GIS data preparation. The surface geological lithology of LXB is extracted from the Chinese stratigraphic lithology dataset (1:2,500,000), which mainly includes geological lithology,

geological body boundary, amphibole schist, crater point and other elements. And we measured the river bed levels and groundwater levels of the Quaternary aquifer and carbonate aquifer along the Langxi River, and the measurement locations are shown in Figure 1, and the time series of the average groundwater level in LRB are shown in Supplementary Figure 1. The maximum root depth of the watershed vegetation was drawn based on the effective soil depth and effective soil volume related to presence of gravel and stoniness in the Harmonized World Soil Database and field surveys. Based on the measured

groundwater level contour and the water level of the Langxi River, we can draw the surplus and deficit reaches where the groundwater recharges river. Combined with the DEM topographic map, the groundwater confined artesian area can be divided, and the groundwater level in this area can be used to determine the GDEs distribution area with shallow groundwater. In addition, water hydrochemical and groundwater fauna sampling also belong to field survey, see the verifying part of GDEs for details.

## 3 Results

### 3.1 Distribution of Potential GDEs

To further identify water areas, bare land, wetlands, vegetation, and other features using quantitative indices and ultimately determine the potential distribution of GDEs, we compared different division methods including the one proposed by Barron et al. (2014). Supplementary Figure 2 displays the distribution characteristics of NDVI and NDWI in the study





area at the end of the dry and wet seasons. In the central and southern plains of the study area, NDVI remained high at the end of the wet season (Supplementary Figure 2a) and slightly decreased at the end of the dry season (Supplementary Figure 2b), indicating that the vegetation in this area primarily experiences rapid drying. In the northern part of the study area, adjacent to the Yellow River, where numerous crops are cultivated, the NDVI value at the end of the dry season ranged between 0.2 and 0.3, slightly higher than the average value of 0.1 (Supplementary Figure 2b). It is noteworthy that, unlike other similar studies, the NDWI did not exhibit significant differentiation at the two time points. The NDWI in the southern mountainous area and the northern planting area ranged between 0.3 and 0.4 at the end of the wet season, while in other areas, it ranged between 0 and 0.2  (Supplementary Figure 2c). At the end of the dry season, the overall NDWI in the study area improved, but the spatial distinction was even less obvious (Supplementary Figure 2d).

Figure 5a and 5b illustrate the change rates of WET and NDBSI in the wet and dry seasons between 2020 and 2021. The purple boundary line in the figure represents the shallow underground level identified by the GDEs distribution area identification method. The WET index shows a higher change rate in the plain area compared to the hilly area, where the humidity does not change much. The plain area near the boundary between the hills and the plain experiences a significant change rate in the WET index. Conversely, changes in the WET index are relatively small along the river banks and in the northern part of the lower-lying basin (Figure 5a). In contrast to the WET index, the NDBSI changes relatively insignificantly in the plain area but exhibits larger changes in the ridges of the basin runoff area (Figure 5b).

The average difference between WET and NDBSI during the dry and wet seasons indicates the variation in water availability between the two seasons with different amounts of water (Figure 5c). High differences in certain areas suggest the availability of relatively stable water supply during the dry season. Analysis of lithology and water sources in the basin reveals that karst groundwater supply is the major source of stable water supply in these areas. Eq.7 is used to estimate the potential distribution range of GDEs based on remote sensing images during the dry and wet seasons of the same year, as depicted in Figure 5d. The potential GDEs range identified from the overlay map of the dry season reveals that, although the hilly area has abundant vegetation, the soil moisture is poor (Figure 5a and 5b), and thus, these areas may not be considered as potential GDEs distribution areas (Figure 5d green area). However, the variation trends of WET and NDBSI indices can be well distinguished in the plain area, which is the boundary between the mountainous and non-mountainous regions. As a result, we can extract potential GDEs areas with better soil moisture stability throughout the year (Figure 5d pink area). The distribution area of potential GDEs is relatively extensive, covering the upper and lower reaches of rivers, as well as some vegetation areas in hilly plains. Nevertheless, further hydrogeological investigations are required to determine if these areas can receive groundwater recharge.

The charts in Figure 6 demonstrate that the difference index, NDVI, and the centroid scatter of NDWI can effectively distinguish between vegetation within and outside the potential GDE range. Furthermore, the permanent water body in the area remains relatively consistent between the dry and wet seasons, aligning closely with the 1:1 line. The discernment between vegetation and water is highly accurate, with the slow drying vegetation exhibiting a difference index range of -0.4 to 0 at the end of the wet season and 0 to 0.4 at the end of the dry season. The difference index of fast-drying vegetation in





the plain area ranges from 0.4 to 0.8 at the end of the wet season, and from -0.4 to 0 at the end of the dry season. In contrast,
the difference index between fast-drying vegetation and crops in mountainous areas is in the range of 0.4 to 0.8 at the end of
the wet season, and in the range of 0.6 to 1 at the end of the dry season. It is noteworthy that, compared with crops, the same
difference index at the end of the wet season is smaller for the fast-drying vegetation in mountainous areas at the end of the
dry season, indicating a smaller regression coefficient between the two (Figure 6a).

    Numerous studies define the NDVI range of 0 to 0.4 as the area covered by sparse vegetations, while areas with NDVI
greater than 0.4 are considered as the area covered by vegetations. The most significant difference is observed between the
NDVI of crops during the wet and dry seasons and the vegetation in the basin. The NDVI of slow-drying vegetation at the
end of the dry season is higher than that of fast-drying vegetation, and its regression coefficients at the end of the wet and dry
seasons are also greater than those of fast-drying vegetation (Figure 6b). These findings are consistent with the research by
Barron et al. (2014) on distinguishing between slow-drying and fast-drying vegetation in Western Australia. Compared to the
previous two indices, NDWI is not effective in distinguishing the range of vegetation, including GDEs, due to various
factors such as geographical location and climate (Figure 6c). This indicates that the selected remote sensing index in the
study has a relatively strong feasibility.

### 3.2 karst springs

    Based on historical data and hydrogeological survey results, the springs in the basin are mainly composed of ascending
springs, depression springs, sinkholes, groundwater seeps, and artificial artesian wells. Table 1 displays the location, type of
spring, flow rate, and other relevant information of the karst spring in the basin.

    In the Hongfanchi Town area, the majority of the Zhangxia Formation aquifers are exposed on the surface, with the
buried limestone ranging from 5 to 60 meters from south to north. Underlying the rock strata is a purple shale and mudstone
of the Mantou Group, which acts as a superior water-resisting layer compared to Zhangxia Group aquifers in other regions.
The bottom of the Zhangxia Group limestone, along the karstic fissures, is where tectonic development occurs, and is mainly
exposed in the form of depression or descending springs, such as the Shuyuan spring, Ding spring, Bajian spring, among
others. Among these springs, Shuyuan Spring has the highest daily spring discharge of over 9,700 $m^3$ $d^{-1}$.

    Along the left bank of the Langxi River, some springs, such as Longchi and the artesian wells, belong to the ascending
springs category. Longchi spring water is pressurized in the confined karst aquifer, flowing upward, and gushes out with an
average discharge of 635$m^3$ $d^{-1}$ through the thin Quaternary strata, which consists of pebbles mixed with sandy clays.

    Another significant spring in the basin is Huquan Spring, which belongs to the sinkhole and has an average discharge of
835.6$m^3$ $d^{-1}$. It exposes a large amount of water to the surface, especially during periods of abundant rainfall. The basin also
contains other karst springs and groundwater seeps that are scattered throughout the area. Some of these springs flow
seasonally, while others have ceased flowing altogether. Additionally, some of these springs have emerged as a result of
water engineering construction and groundwater extraction.





### 3.3 S-GDEs and V-GDEs

Based on GIS data and hydrogeological surveys, we acquired information on the regional maximum root depth, river bottom, and groundwater level, which were necessary to identify S-GDEs and V-GDEs (Figure 7a). Additionally, we qualitatively assessed the gaining and losing river segments (Figure 7b) and examined the geological profile of a typical descending spring in the LRB (Figure 2).

The analysis of underground water table depths reveals that the shallow water table area (0 to 5m) is primarily located in the middle of the watershed where the tributaries converge, and numerous karst springs are situated nearby. The vegetation in LRB is predominantly composed of deciduous broad-leaved forest and deciduous open shrubs, with relatively developed underground root systems owing to the year-round flow of rivers. The maximum root depth in the basin ranges from 1.5 to 2.5 m, with some exceeding 4.5 m. Areas with deeper root depths are present on both sides of the river. In the lower reaches of the Langxi River near the Yellow River, the roots of vegetation are relatively shallow, consistent with the spatial distribution of surface lithology. However, the reason for the shallow roots is not caused by the surface lithology. In fact, it is due to the abundance of water in the area, and a large number of farmlands are distributed here. The riverbed bottom level changes more gently from upstream to downstream compared to the change in elevation.

### 3.4 Distribution of GDEs in LRB

According to the four diagnostic criteria, we subdivided the potential GDEs into S-GDEs and V-GDEs, and marked the K-GDEs with their site environmental characteristics. This allowed us to obtain the distribution area of GDEs in the basin, as shown in Figure 8. The three colors represent three different types of GDEs, and the same color system with different brightness represents the gradient spatial kernel density of the GDEs. The GDEs in the basin are mainly located in the central and western parts of the LRB, covering an area of approximately 49 km$^2$, which accounts for 29% of the basin's total area. Although the river runs through the karst area, the spatial distribution of its center is not consistent with the surface water-enriched river course and small reservoirs, which is a notable characteristic of GDEs identified in this study. The surface water in the karst region recharges the groundwater through lateral discharge, but this recharge is not concentrated on the river channel; instead, both sides of the river channel are equally affected. Therefore, the shape of GDEs reflects the trend of underground aquifers on the ground, which is in agreement with the GDEs definition and distribution results reported by Erostate et al. (2020) and Duran-Llacer et al. (2022). We divided the coverage of GDEs into four levels based on the kernel density gradient histogram, which aligns better with the actual distribution under different water recharge conditions.

The results depicted in Figure 8 indicate that the distribution of GDEs is significantly impacted by human activities, not only in the surface water system but also in the groundwater system, including aquifers. It is evident that the GDEs in the northern and eastern regions terminate at the dam, indicating that water conservancy facilities situated on the main channel of the Langxi River obstruct and disrupt the connectivity of surface and underground water.





### 3.5 Ecohydrological signals of GDEs

#### 3.5.1 Base flow

Based on the hydrogeological graph spanning from 1990 to 2015 (Figure 9), it is evident that the river flow near the
Shuyuan spring section is highly dependent on precipitation. The maximum discharge recorded was 1450 L s$^{-1}$ in 2004,
while the minimum average discharge was 6.87 L s$^{-1}$ in 2015, which is more than 211 times lower than the maximum value
(Figure 9a). The average annual discharge and daily discharge are 89.66 L s$^{-1}$ and 7746 m$^3$, respectively. Despite this
variation in flow, the base flow is maintained even during dry seasons, with the base flow index (BFI) accounting for
54.15% of the river flow. Among them, the minimum BFI was 0.369 in 1991, and the maximum was 0.845 in 2013. It can be
seen that the base flow has always made a large contribution to the runoff of the basin. Even in the long period of no-rain
recharge, the base flow presents a decay trend and lasts for a long time until the next rainfall. Therefore, we chose July 1993
to July 1994 as a typical year to analyze the change process of precipitation and Shuyuan section discharge to illustrate how
groundwater and springs feed the river. It can be seen that there were two obvious flowing decline processes in this year
(Figure 9b). Among them, there was almost no precipitation between November 1993 and April 1994, indicating that the
river flow was mainly influenced by groundwater and spring discharge.

We selected the hydrological graph from November 1993 to April 1994 (Figure 10c) during a period with no
precipitation and groundwater exploitation. It was found that the rivers during this no-rain recharge period were mainly
sustained by spring water, and the flow rate showed an exponential decay trend. The flow attenuation presents three stages,
which we believe are represented by the red line segment indicating the concentrated flow with turbulence in the karst
conduit to recharge the river; the black line indicating the large corrosion voids and fractures supplying fracture flow; and
the blue lines indicating the small corrosion cracks, fractures, and intergranular pores recharging the diffuse flow aquifer
with diffuse flows and laminarity.

Upon comparing the attenuation coefficient ($\alpha$) of different stages, it is evident that the attenuation coefficient of the
first sub-dynamic stage (0 to 10 days) is the largest, which is 0.0985. The attenuation coefficient of the second sub-dynamic
stage (10 to 51 days) has significantly weakened, while the attenuation coefficient of the third sub-dynamic stage (51 to 141
days) gradually approaches 0. From the above analysis, it is evident that even during a prolonged period of no rain recharge;
the flow attenuation can be sustained for an extended period, indicating that groundwater has consistently contributed to the
Langxi River's flow.

#### 3.5.2 Hydrochemical types

In this paper, ten water samples, including karst groundwater, Quaternary pore water, and surface water, were collected
and tested in the laboratory. The results were then plotted onto a piper plot, as shown in Figure 10a. The chemical indicators
of the water, such as DO, pH, EC, and total dissolved solids (TDS), were analyzed and clustered to create a diagram, which
is presented in Figure 10b.





Based on the hydrochemical analysis, most of the water samples had a pH value in the range of 7 to 7.5, indicating a
weakly alkaline environment. The hydrochemical type of the limestone aquifer water samples in the monitoring well was
mainly $HCO_3$-Ca·Na, while the other water samples were of the $HCO_3$-Ca type. From the piper plot and the cluster diagram,
the data can be divided into three clustering groups. The first clustering group consists of groundwater with depths ranging
from 60 to 90 m in the monitoring well, reflecting the groundwater characteristics of the Cambrian Zhangxia Group karst
confined aquifer. The second clustering group includes water from Longchi, Hu spring, Shuyuan spring, Ding spring, and
Bajian spring, which are mainly natural outcrops of groundwater, reflecting the characteristics of karst spring water. The
third clustering group consists of the Quaternary pore water and water from the Huiquan reservoir, reflecting the reservoir
mainly replenished by shallow groundwater. In fact, the cluster partitioning is not particularly clear, especially for the
descending springs like Shuyuan spring. These springs are not only closely related to the karst aquifer but also have a good
hydraulic connection with the Quaternary aquifer. On the other hand, the Huiquan reservoir is a surface reservoir impounded
by river barrages and receives a lot of groundwater recharge. As a result, it exhibits similar water chemical characteristics to
spring water and Quaternary pore water. Therefore, it can be observed that the interaction between surface water and
groundwater in this region is relatively strong, and the water chemical characteristics of GDEs have obvious differences but
also some similarities.

**3.6 Groundwater fauna species**

Samples of groundwater fauna were collected from various types of GDEs including karst cave (sink hole), karst
aquifer, depression spring, ascending spring and river hyporheic zone. Based on laboratory identification, the primary
stygofauna found in these GDEs were *Neocaridina denticulata sinensis*, *Chironomid larvae*, *Petopia.sp*, *Dytiscidae*,
*pelopia*, *Radix lagotis*, *Gyraulus*, *Galba pervia* (Figure 11; Table 2).

Finding groundwater fauna near karst caves is easy because there is an abundance of food, making it easier for them to
survive. Hu Spring, a natural karst cave, is home to three species of *Chironomid larvae*; *Anisogammarus sp*.; *Radix lagotis*
were found in it. Typically, these faunas mainly comprise of *Arthropods*, *Coelenterates* and *Mollusks* that live in
groundwater throughout their entire life cycle, and are known as stygobites, a true groundwater fauna.

As groundwater levels deepen, there are fewer aquatic organisms present. Hahn and Matzke (2005) discovered that the
screen of an artificial borehole would not prevent groundwater organisms from passing through, but we did not observe any
organisms in the two new monitoring wells. Many species of groundwater fauna, including *Neocaridina denticulata
sinensis*, *Chironmidae larvae and Dytiscidae*, were found in the sink pool of depression and ascending springs.
*Neocaridina denticulata sinensis*, also known as *Penaeus monodon*, is a flagship species among the many aquatic organisms
found in the sink pool. It has a dark green body and is a very small shrimp, measuring only 5 to 10 mm in length. It primarily
inhabits freshwater ponds with abundant aquatic plants, and has the highest yield in autumn. There are also many species
distributed throughout the river hyporheic zone. The river hyporheic zone also harbors a variety of species. *Dytiscidae*, also





known as *Terrapin* or *Aquatic beetles*, are a type of *Arthropod* that range from 3 to 5 mm in length, with varying individual sizes. The adults have a long streamlined body, flat and smooth with an arched back, and developed bristles. *Radix lagotis*, a type of aquatic snail, is about 1.5 mm long and has a thin, slightly hard shell with an elliptical shape. It is found in the wild. *Gyraulus*, another type of aquatic snail, measures about 8 mm in length. *Galba pervia*, a species of freshwater snail, is about

4 mm long and can be found in various still water and slow-flowing water. These species are representative of benthic types of groundwater fauna. *Chironomidae* is a widely distributed species among groundwater fauna, with *Chironomid larvae* found particularly in water bodies that have good quality and high dissolved oxygen (DO) content. This species belongs to the stygophilies, which is a type of groundwater fauna.

In summary, various types of GDEs such as karst caves, karst aquifers, karst springs, and river hyporheic zones, host

distinct populations of groundwater fauna and iconic species. This information can be used to confirm the distribution of GDEs.

## 4. Discussion

### 4.1 Four diagnostic criteria framework

The aim of the framework presented in this paper is to offer a comprehensive methodology, rather than a specific

method, for identifying and evaluating GDEs. This methodology not only incorporates existing approaches for identifying GDEs but also applies to basins with varying climatic and geological conditions. For instance, the surface vegetation type can be more accurately identified using NDVI and NDWI in Barron et al. (2014)'s study, whereas in our framework, it is advisable to experiment with multiple indices. The outcomes shown in Figure 6 additionally confirm the viability of using remote sensing indices to identify potential GDEs, as well as the unsuitability of NDWI in LRB.

The kernel density function chosen to be used in our framework is to expand the areal scale by using the probability density for the uncertain groundwater recharge range (Pérez Hoyos et al., 2016). The advantage of this is that the GDEs area can be quickly identified using simple survey data under conditions determined by factors such as topography and remote sensing (Doody et al., 2017; Paz et al., 2017). It is important to acknowledge that the methodology for identifying GDEs still faces limitations, particularly in the precise definition of the scope of influence of fine-grained GDEs (Martínez-Santos et al.,

2021), such as K-GDEs. This challenge remains unresolved in the field. Even for S-GDEs and V-GDEs, determining their scope requires extensive on-site investigation and hydrogeological, groundwater, and vegetation data (Erostate et al., 2020). Furthermore, these data must be relatively time-sensitive, as per existing research.

### 4.2 Groundwater connectivity should be focused

The four diagnostic criteria proposed in the study that combines GIS and hydrogeological survey are valuable for the

identification and mapping of GDEs. However, remote sensing methods can only identify potential distribution areas of GDEs, and they are not sufficient for accurately describing the actual distribution of GDEs. For instance, we need to



determine when and how much groundwater replenishes GDEs, as well as the extent to which the roots of various vegetation communities can absorb groundwater. These groundwater connectivity processes necessitate careful hydrogeological investigation, particularly through the use of hydrogen and oxygen isotope tracing methods. Given that identifying GDEs is a

fundamental issue, some authors have also examined the impact of groundwater extraction and other factors (Gou et al., 2015; Münch and Conrad, 2007; Pérez Hoyos et al., 2016).

### 4.3 Groundwater fauna tracing would be necessary

The relative spatial independence of groundwater fauna in karst spring-type GDEs can serve as an indicator for identifying such systems. However, sampling stygofauna poses a greater challenge due to the complex trajectories of

groundwater fauna, including species fluctuations observed in sink holes, as well as limitations in sampling and tracing methods. Various methods are currently used for sampling stygofauna, each with its own advantages and disadvantages (Hahn, 2002; Leijs et al., 2009; Smith et al., 2016). Nevertheless, there are currently no standardized methods for sampling stygofauna (Hahn and Matzke, 2005). Net sampling is the most efficient and cost-effective method for quickly obtaining numerous stygofauna samples. However, using a mesh size of 74 μm or even 40 μm makes it challenging to collect all

sample types, particularly during dry seasons. Water sampling, which involves pumping with a homemade bio-pump and filtering with a 74 μm filter, is another swift and economical option, but the pumped water volume is limited. In some cases, columnar species were not discovered due to screening barriers, making it challenging to track groundwater fauna. To address this, real-time monitoring, tracing, and DNA sequencing technologies can extract essential genetic markers of groundwater fauna, facilitating the study of biotic connectivity between groundwater and GDEs. This could be a significant

breakthrough in groundwater fauna research.

### 4.4 Impacts of human activities and climate change should be more concerned

GDEs have always been impacted by climatic variations, but with the increasing scale of human activities, new stresses are emerging. It is essential to focus on the hydrological and ecosystem response to these stresses. To better understand the vulnerability of GDEs, regional studies are necessary. It is crucial to comprehend the role of human activities and climate

change to identify potential human impacts and put climate change trends into perspective (Barron et al., 2012; Gurdak et al., 2007; Kløve et al., 2014; Liu et al., 2016; Randhir and Hawes, 2009). Karst groundwater ecosystems are generally more vulnerable than other ecosystems due to the specific features of the karst zone, such as high permeability, rapid infiltration or recharge rates, and being mainly controlled by karst conduits. Therefore, it is essential to monitor changes in the static water level, groundwater discharge to streams, and riparian plants resulting from changes in precipitation, runoff, and water use.

Additionally, we must determine if land consolidation, engineering construction, and other human activities are disrupting the groundwater upflowing or recharging route. To fully comprehend these complex factors and interactions, further studies are necessary (Kløve et al., 2011).





## 5. Conclusion

This paper proposes a framework of four diagnostic criteria that combines remote sensing, GIS data, and field hydrogeological surveys. This framework can effectively identify and map different types of GDEs in a typical karst basin. Compared to the traditional NDVI and NDWI index division, the difference index of WET and NDBSI has better adaptability for identifying potential GDEs distributions. The GDEs are then mapped using spatial kernel density functions. The results reveal that there are three main types of GDEs. River-type and vegetation-type GDEs are concentrated along the Langxi river riparian zone, while karst aquifer-type GDEs are scattered throughout the basin. Each type of GDEs displays special ecohydrological signals. For instance, one of the obvious signals for the gaining stream is the base flow index, which can reach about 54.15%, keeping the river flowing even during extremely dry seasons. The second signal is the clustering of hydrochemical characteristic ions, which reveals whether the GDEs are replenished by karst groundwater or other water sources. The third unique ecohydrological signal is the groundwater fauna that lives in different types of GDEs. These three signals can be utilized to evaluate the accuracy of identifying and mapping GDEs. However, the knowledge gap regarding the ecohydrological connectivity between groundwater and GDE can be improved by utilizing isotope analysis, stygofauna tracing, and DNA sequencing technology under the recommended four-diagnostic criteria framework in the future.

## Acknowledgements

This work was supported by the National Natural Science Foundation of China (No. 41572242), the National Key Research & Development Program of China (No. 2018YFC0408002), and the Provincial Water-conservancy Scientific Research and Technology Extension Project of Shandong province (No. SDSLKY201703). We are grateful to Dr. Ana Sofia Reboleira from University of Copenhagen for helping stygofauna sampling in field.

## Author contribution

M.L. and F.L. developed the initial and final versions of this manuscript and analyzed the data. D.F., H.C., K.W., X.C., and J.H. contributed their expertise and insights to oversee the analysis.

## Competing interests

There's no conflict of Interest in this paper.

## Code/Data availability

The data that support the findings of this study are available from the corresponding author upon reasonable request. The process code of GDEs identification and mapping in this research can be downloaded at https://code.earthengine.google.com/ee0675e5501a03235024f4bb3b941992.



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



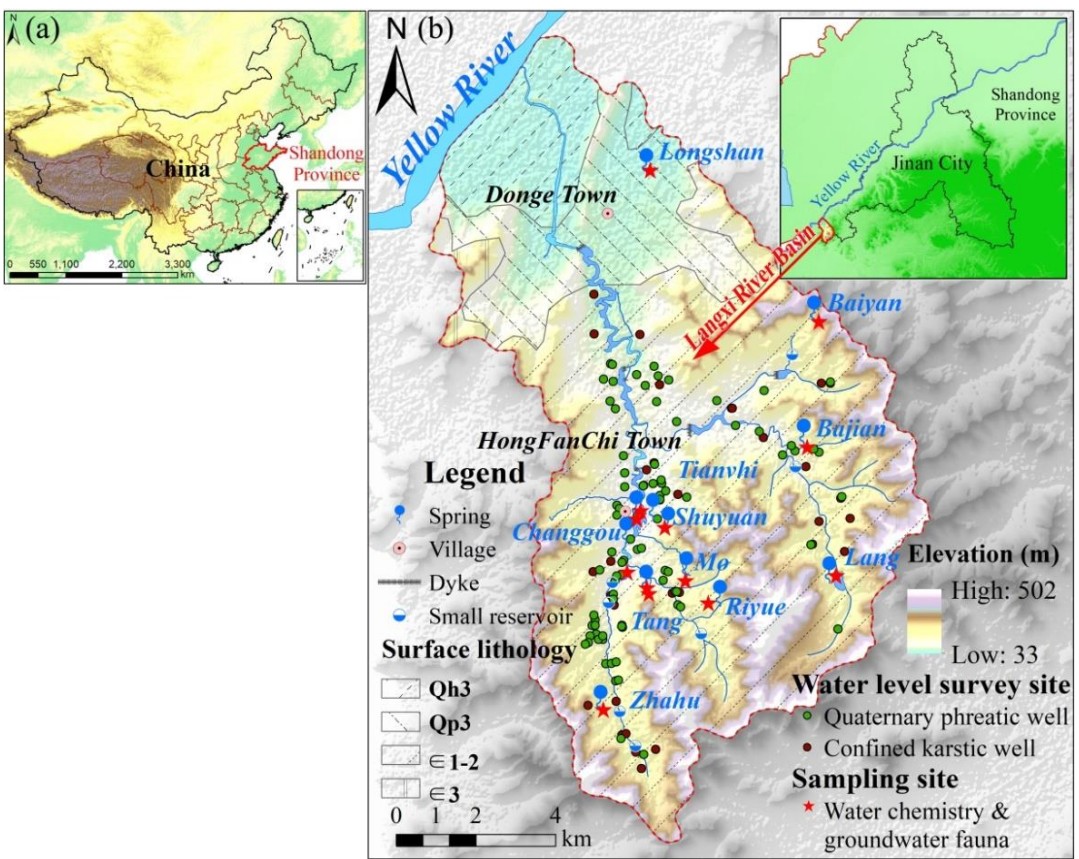


**Figure 1: (a) The location of Shandong Province, China; (b) The location, lithology, topography, spring water, groundwater level survey points, hydrochemical groundwater biological sampling points of the Langxi River Basin.**

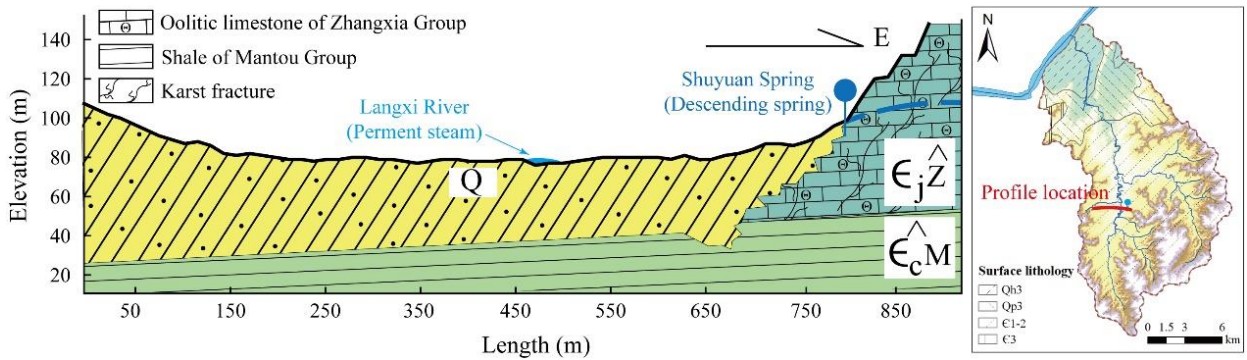

**Figure 2: Hydrogeological profile of Shuyuan spring in LRB. The dotted line shows the characteristics of the water table in the**
**geological section. The geological types in the figure are Qh3: Holocene fluvial alluvial deposits; Qp3: Upper Pleistocene gravel layer; €1-2: Cambrian Zhushadong-Zhangxia Formation limestone; €3: Cambrian Gushan-Chaomidian Formation limestone.**



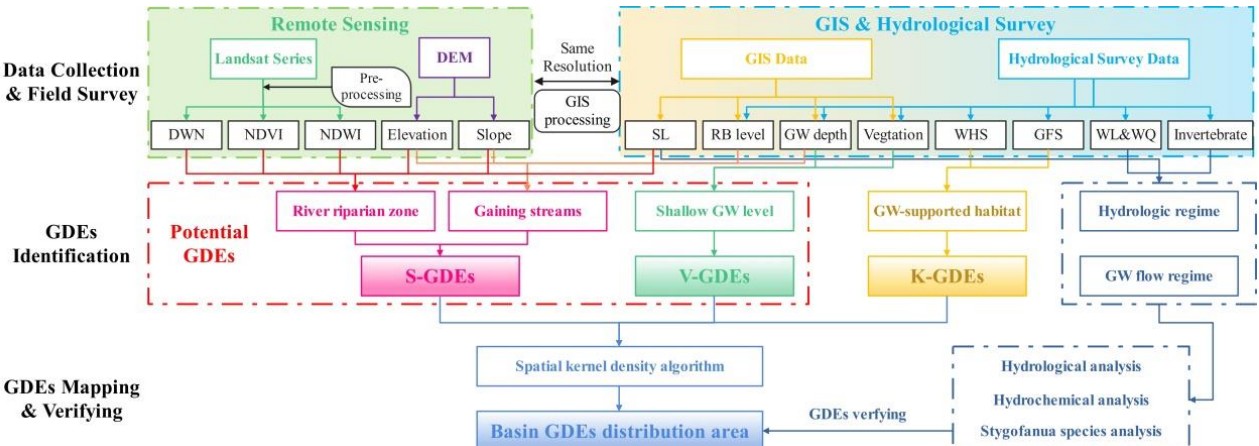

**Figure 3: Diagnostic framework for GDEs identification, mapping and verifying. Note: DEM: digital elevation model; DWN: the difference between wet index and the normalized difference built-up and soil index; GIS: geographic information system; SL: surface lithology; RB: river bed; GW: groundwater; WHS: water hydrochemical sampling; GFS: groundwater fauna sampling; WL & WQ: water level and water quality; S-GDEs: stream-type GDEs; V-GDEs: vegetation-type GDEs; K-GDEs: karst-type GDEs.**

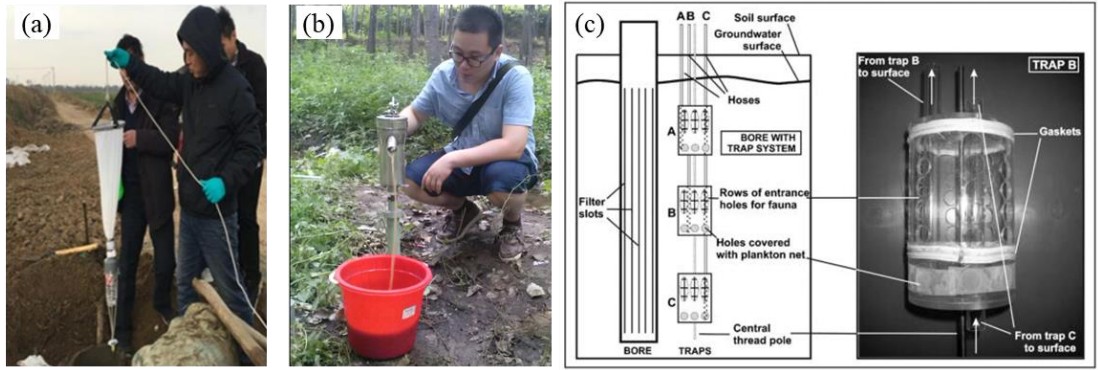

**Figure 4: Stygofauna sampling combining (a) phreatobiological net sampling, (b) micro-pump sampling and (c) unbaited trap sampler (Hahn, 2005).**



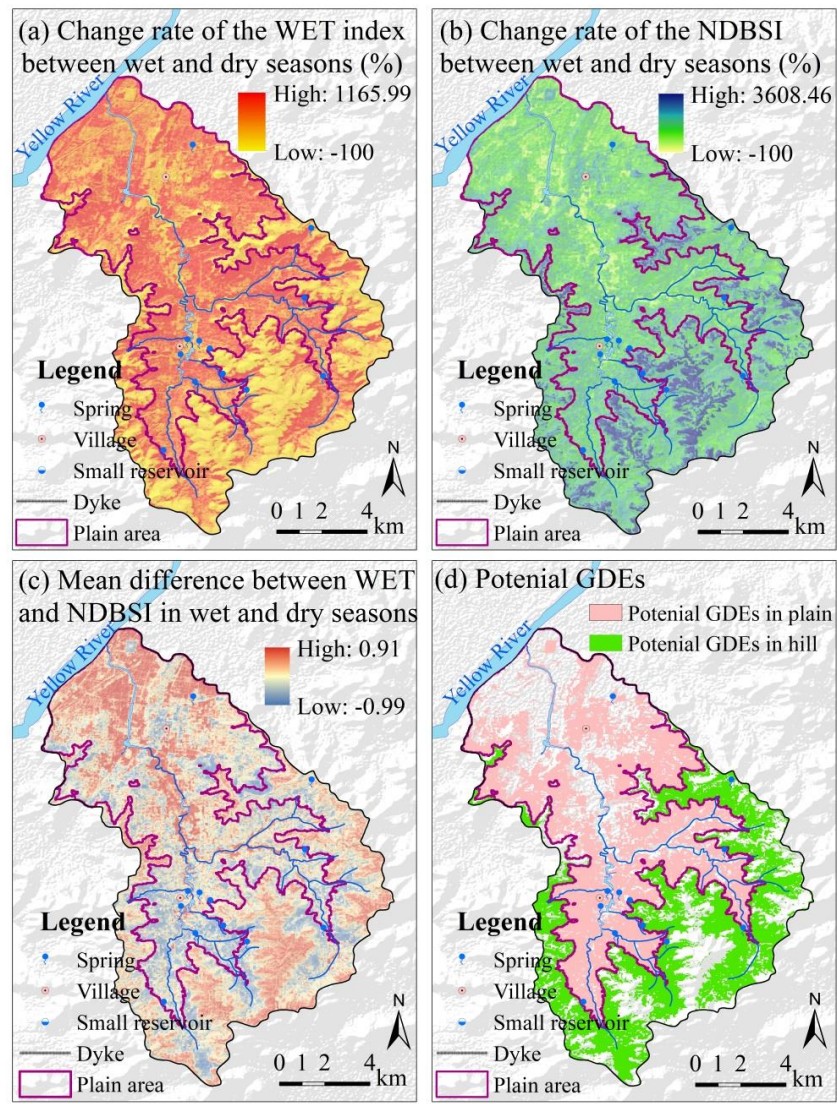

**Figure 5: Change rate of WET (a) and NDBSI (b) between wet and dry season in 2020 to 2021, (c) the mean difference between WET and NDBSI, and (d) the potential GDEs in plain area and basin using remote sensing data.**


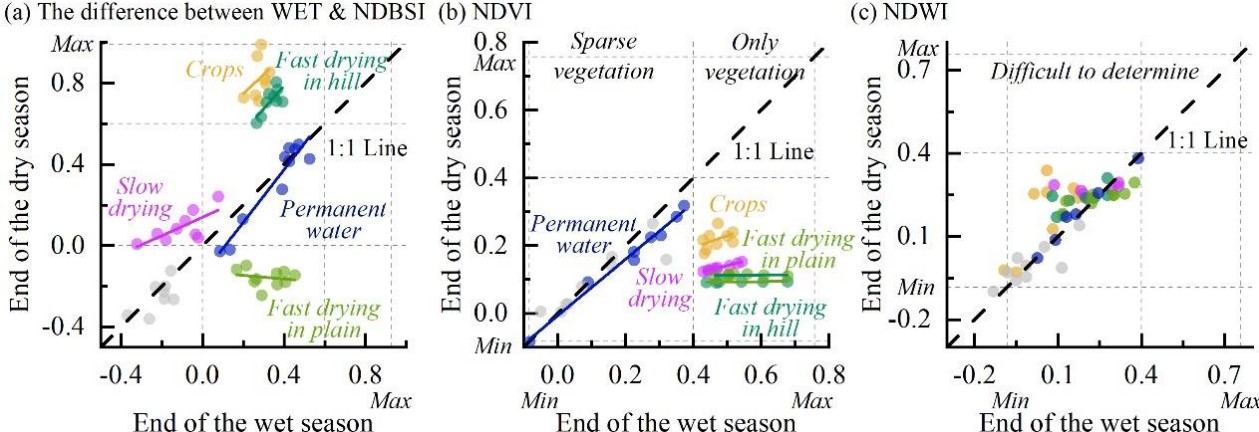

**730**   **Figure 6: The centroid scatterplots for the difference between WET and NDBSI (a), NDVI (b) and NDWI (c) in the end of the wet season and the dry season (2020 to 2021).**

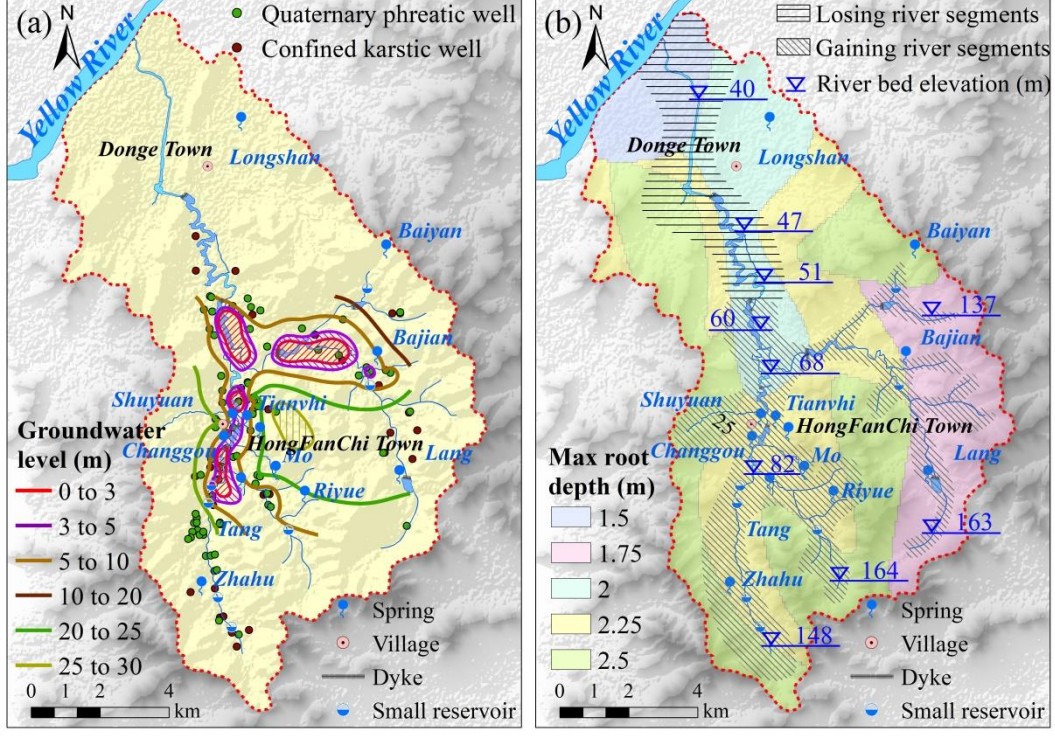

**Figure 7: Summary of hydrogeological survey data in LRB. (a) Distribution map of groundwater level, (b) Maximum vegetation root depth, river bed elevation, and water gain and loss in the river segments.**






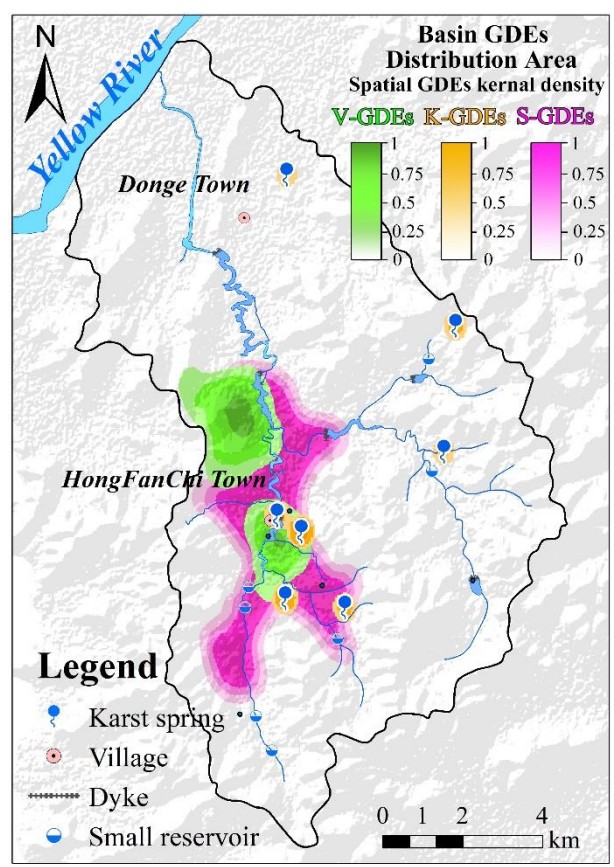

**Figure 8: Langxi River Basin GDEs distribution area and their spatial GDEs kernel density.**





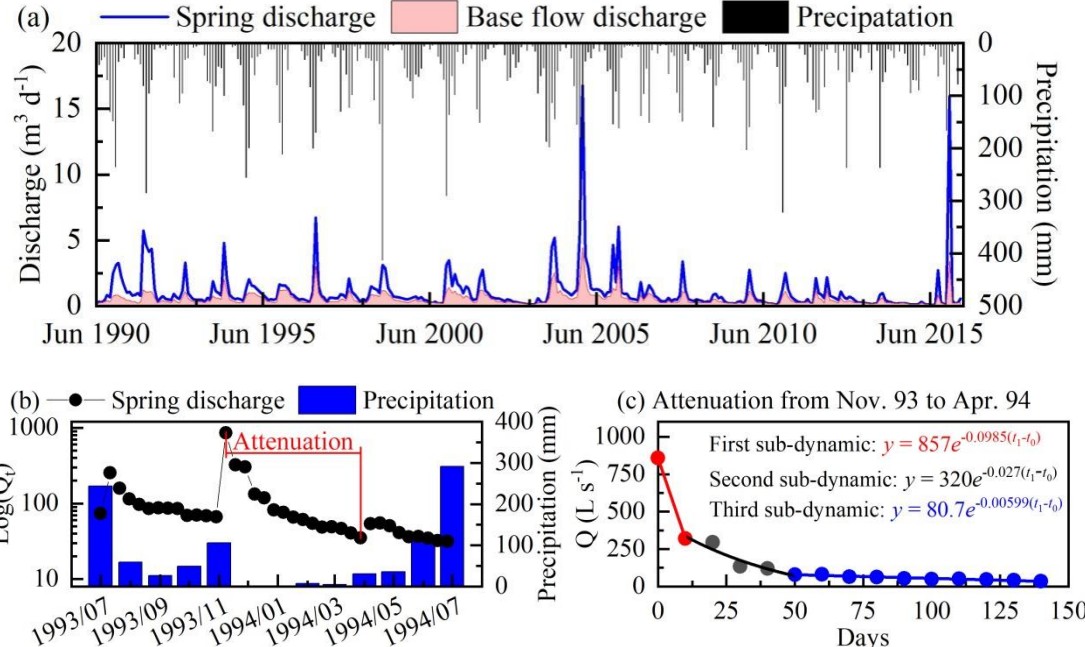


**Figure 9: (a) Relationship between precipitation and Shuyuan section river flow and its base flow from 1990 to 2015; (b) Hydrologic graph of Shuyuan spring from July 1993 to July 1994; (c) Spring discharge attenuation curve of Shuyuan spring from November 1993 to April 1994.**



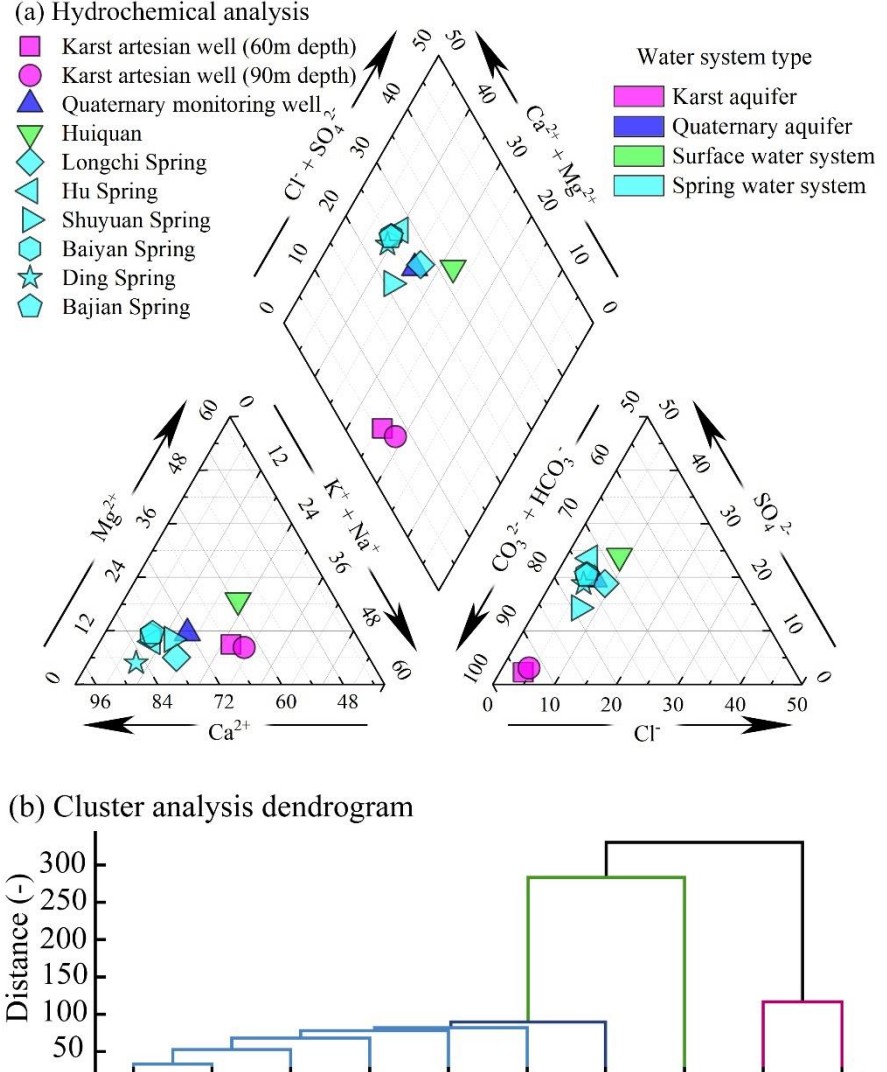


Figure 10: Hydrochemical characteristics of ten water samples in LRB.

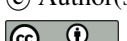



| (a) | *Galba.jervia.sp* | |
|---|---|---|
| Length: 0.7 mm | Phylum | Mollusca |
| | Class | Gastropoda |
| | Order | - |
| | Family | Lymnaeidae |
| | Genus | *Galba* |

| (b) | *Radix lagotis Schrank* | |
|---|---|---|
| Length: 2 mm | Phylum | Mollusca |
| | Class | Gastropoda |
| | Order | - |
| | Family | Lymnaeidae |
| | Genus | *Radix* |

| (c) | *Chironmidae* | |
|---|---|---|
| Length: 3 mm | Phylum | Arthropoda |
| | Class | Insecta |
| | Order | Diptera Nematocera |
| | Family | Culicomorpha |
| | Genus | *Chironomoidea* |

| (d) | *Gyraulus.sp* | |
|---|---|---|
| Length: 1.5 mm | Phylum | Mollusca |
| | Class | Gastropoda |
| | Order | - |
| | Family | Planorbidae |
| | Genus | *Gyraulus* |

| (e) | *Dytiscidae.sp* | |
|---|---|---|
| Length: 0.8 mm | Phylum | Arthropoda |
| | Class | Insecta |
| | Order | Coleoptera |
| | Family | Dytiscidae |
| | Genus | *Cybister* |

| (f) | *Anisogammarus.sp* | |
|---|---|---|
| Length: 3 mm | Phylum | Arthropoda |
| | Class | Malacostraca |
| | Order | Amphipoda |
| | Family | Anisogammaridae |
| | Genus | *Anisogammarus* |

**Figure 11: Images of some groundwater fauna samples. (a)** *Galba.jervia.sp***; (b)** *Radix lagotis Schrank***; (c)** *Chironmidae***; (d)**
*Gyraulus.sp***; (e)** *Dytiscidae.sp***; (f)** *Anisogammarus.sp***.**

**Table 1: Summary of karst spring's field investigation in Langxi River Basin (LRB).**

| Spring name | Genetic type | Discharge (m³/d) | Hydrochemical / hydrobiological sampling | Supplementary Description |
|---|---|---|---|---|
| Longchi spring | Ascending spring | 635.0 | Hydrobiological | Not dry all year round |
| Shuyuan spring | Erosion descending spring | 9787.4 | Both | Not dry all year round |
| Hu spring | Sink hole | 835.6 | Both | Flow cutoff in dry years, can be pumped |
| Riyue spring | Erosion descending spring | 54.8 | Both | Not dry all year round, the spring flux is small |
| Ding spring | Erosion descending spring | 939.7 | Hydrobiological | Flow cutoff in dry years |
| Baiyan spring | Erosion descending spring | 575.3 | Both | Flow cutoff in dry years |
| Bajian spring | Erosion descending spring | 402.7 | Both | Flow cutoff in dry years |
| Tianvhi spring | Difficult to determine | - | Hydrobiological | No water gushing |
| Changgou spring | Erosion descending spring | - | Hydrobiological | Be buried by landslides |
| Zhahu spring | Erosion descending spring | - | Both | Overflow in flood season, seasonal spring |
| Lang spring | Erosion descending spring | - | Both | Plunge down to the reservoir |
| Longshan spring | Erosion descending spring | 367.1 | Both | Overflow in flood season, seasonal spring |





**Table 2: Summary of groundwater fauna sampling of the GDEs.**

| Sample No. | Sample site | GDEs type | Recharge water source | Groundwater fauna species | Note |
|---|---|---|---|---|---|
| HF-1 | Karst artesian well (60m depth) | K-GDEs | KGW | *No Arthropods* | Karst aquifer, new monitoring well |
| HF-2 | Karst artesian well (90m depth) | K-GDEs | KGW | *No Arthropods* | Karst aquifer, new monitoring well |
| HF-10 | Shuyuan spring | K-GDEs | KGW and QPW | *Neocaridina denticulata sinensis* | Depression spring discharges into the river with a lot of moss |
| HF-11 | Hu spring (sink hole) | K-GDEs | KGW | *Chironomid larvae; Anisogammarus sp.; Radix lagotis* | Karst cave |
| HF-12 | Mochi spring | *S-GDEs* | KGW | *Dytiscidae, Neocaridina denticulata sinensis, Chironomid larvae* | Karst spring pond with 5 m depth and the water is as colored as ink, and the water contains a large amount of Spirogyra. |
| HF-13 | Long spring | K-GDEs | KGW and part of QPW | *Neocaridina denticulata sinensis, Chironomid larvae* | Ascending spring |
| HF-16 | Langxi river (downstream section) | *S-GDEs* | *SW and part of* KGW | *Petopia.sp, Radix lagotis, Petopia, Gyraulus, Galba.jervia.sp* | Hyporheic zone, the river channel is 80~90m width and the water flow is slow, containing a large number of algae. |

**Note:** K-GDEs and S-GDEs are karst-type GDEs and stream-type GDEs; KGW, QPW, and SW are karst groundwater,

Quaternary pore water, and surface water.