# Peer review of "● Supplementary Material"

_Hydrology and Earth System Sciences, 2023_

## Referee Comment (RC2)

The manuscript by Li et al. focuses on the identification and analysis of groundwater-dependent ecosystems (GDEs) in a specific region. The authors propose a four-diagnostic criteria framework for identifying GDEs based on remote sensing, GIS data dredging, and hydrogeological surveys. Overall, I think this work will be of interest to the HESS readership but needs some reworking to be considered further for publication. I make a few suggestions for improvement below:

1. The result section lacks clarity and structure. It resembles a report rather than a concise summary of findings. Some materials in the results, particularly those pertaining to the authors' methodologies or rationales, such as those indicated on Page 11 Lines 337-339, Page 14 Lines 407-408, and Page 14 Lines 421-423, should be relocated to the method section to enhance organization and coherence.

2. The current version lacks substantial supporting information on ecohydrological signals aiding in mapping GDEs. Section 3.4 primarily focuses on the spatial distribution of GDEs within the study area. Sections 3.5 and 3.6 delve into detailed descriptions of hydro-biogeochemical features observed in GDEs. However, there appears to be a weak connection between these descriptions and the verification of various GDEs, as depicted in Figure 2. Strengthening this connection is essential for a more cohesive presentation of the study's findings regarding the role of ecohydrological signals in GDE mapping.

3. The proposed framework should be compared to other existing approaches to highlight its potential advantages and limitations. This discussion is crucial for providing insights into the novelty and effectiveness of the proposed methodology.

**Specific comments**
Page 4, Line 100: Delete ", etc".
Page 6, Line 183: Please clarify the rationale behind selecting a 10-degree angle.
Page 10, Line 285: Delete "2"
Page 16, Line 495: Font size is not consistent.

---

## Referee Comment (RC3)

Dear Editor,

I have had the opportunity to review the manuscript detailing the assessment of Groundwater Dependent Ecosystems (GDEs) using remote sensing data and hydrogeological surveys in the Langxi River Basin. This study provides an insightful exploration of the methods used for identifying and characterizing GDEs, a topic of great environmental importance due to the critical role these ecosystems play in biodiversity conservation and water resource management. The findings are robust and contribute meaningfully to the field, especially in terms of methodological innovations. However, I have several comments that I believe could further refine and enhance the value of the work presented.

1. The manuscript offers a detailed account of the combined use of GIS and hydrogeological data. However, it would benefit from a more explicit explanation of the choice and configuration of the remote sensing technologies and the rationale behind the specific indices selected for analysis. Providing this information would help in replicating the study in other regions or contexts, enhancing its utility for broader application.

2. The results section effectively illustrates the application of NDWI and NDVI indices in identifying GDEs. However, ensuring that all figures and tables consistently reflect the descriptions provided in the text would improve the manuscript's readability and professional appearance. Specifically, ensuring that legends and captions are clear and that graphical representations directly correspond to the described findings is crucial.

3. While the study discusses the settings used within the numerical models and their boundary conditions, linking these directly to either field conditions or the parameters set during laboratory tests could strengthen the study. A detailed comparison would not only validate the model further but also provide clarity on its applicability in real-world scenarios.

4. The conclusion successfully outlines the study's contributions to understanding GDEs. However, integrating specific recommendations for policymakers on managing groundwater extraction in coastal areas could significantly enhance the

manuscript's impact. Practical guidelines based on the study's findings would be invaluable for regional planning and conservation efforts.

5. The manuscript exhibits a commendable level of detail and scientific rigor. However, to elevate the manuscript's clarity and professional presentation, a comprehensive review of typographical errors, figure consistency, and data presentation is essential. For example, some figures and diagrams exhibit inconsistent use of color schemes and font sizes, which could potentially confuse the reader or detract from the data's visual impact. Additionally, inconsistencies in symbol usage and abbreviation definitions across the text and figures were noted. Ensuring that all graphical representations adhere to a uniform style guide would significantly enhance readability and the visual appeal of the manuscript. It is also recommended to verify the accuracy of all legends and captions to ensure they precisely describe the corresponding figures and tables. Addressing these issues will not only refine the presentation but also bolster the manuscript's overall credibility and ease of understanding.

In summary, this manuscript provides important insights into the assessment and management of GDEs using advanced remote sensing and hydrogeological methods. Addressing the points listed above could strengthen the manuscript's impact, making it a significant contribution to environmental science research. I look forward to seeing these enhancements in the revised manuscript.

---

## Author Comment (AC1)

**Responses to the reviewers**

**Title:** Identification, Mapping and Eco-hydrological Signal Analysis for Groundwater-dependent Ecosystems (GDEs) in Langxi River Basin, North China (hess-2023-151)

**Authors:** Mingyang Li, Fulin Li*, Shidong Fu, Huawei Chen, Kairan Wang, Xuequn Chen, Jiwen Huang

**Thanks to the experts giving so much valuable advice, now the revision notes are shown as following.**

**RC1:**

This is a manuscript that deserves praise. It is difficult to concentrate on rich field experiments in the current setting. This study presents a framework for locating and mapping GDEs based on a mix of remote sensing, GIS, and hydrogeological field experiments in terms of research content and methodologies. The study's newly suggested normalized difference built-up and soil index, together with the difference between the wet index, may be used to assess changes in the water loss rate of plants at various stages of growth. After then, the spectrum of possible GDEs is defined by factors including slope, lithology, and elevation. Groundwater levels, river bed bottom elevation, plant root depth, karst springs, etc. were used to further infer different forms of GDEs. In the end, three factors were used to verify GDEs: groundwater biology, water chemical isotopes, and hydrological rhythm. Gaining a deeper understanding of a basin may be achieved by employing a variety of techniques to examine its biological and hydrological features. I would want to share a few small ideas and inquiries with the author.

Answer: Thank you very much for your review and the suggestions of the three experts. We have revised the paper according to each suggestion.

1. The author employed aquatic biology, hydrology, water chemistry, isotopes, and other elements to confirm. These verification findings demonstrate that spring flow and base flow in the basin have a substantial association, that the water chemistry and isotope composition of distinct water bodies vary, and that the water bodies in GDEs have comparatively separate ecosystems. It is still necessary to enhance the geographical representativeness of the aforementioned results. Is it attainable to categorize and validate GDEs in space, despite the fact that they are challenging to locate and define precisely?

Answer: Thank you very much for your comment. We will reply to your questions in the following points:

The Langxi River Basin (LRB) is a typical continuous carbonate-type karst basin in northern China. Generally speaking, this kind of basin is scattered and relatively independent. The small area contains complex topography and groundwater exchange relationships. Therefore, similar research often selects basins with a small area, which is convenient for detailed investigation. The research methods have operational

commonality in similar basins around the world. For example, this study is based on previous methods and aims at the characteristics of the underlying surface of the LRB, trying to propose a more complete research framework and improved methods. In this study, we targeted the classification and identification of GDEs according to the types of GDEs in LRB. Because different types of GDEs are distributed in LRB, including karst springs and hyporheic zones, they are more typical in northern China. In other places, affected by groundwater overexploitation and high-intensity human activities, GDEs have been seriously damaged. Even though large karst springs exist in some places, they do not form an obvious ecosystem. Therefore, first of all, we believe that choosing LRB as a typical research area is representative of the karst areas in northern China.

In this study, we propose a more systematic and comprehensive method compared to current identification and mapping research. We are not simply doing GDE classification research, because the classification is relatively simple and can be identified in the wild. And our research mainly focuses on recognition and mapping.

Based on previous research, we proposed a framework for identifying, mapping and verifying the distribution range of GDEs. This system can be improved and transplanted to other regions. For example, Barron et al. (2014) used NDVI and NDWI to classify potential GDEs in Australia. This is not significantly feasible in LRB, but this does not mean that its idea is not feasible. We selected appropriate remote sensing indices NDBSI and NDVI and jointly identified potential GDEs in LRB, and the results showed that the discrimination effect was very good. Therefore, we believe that this study is more focused on proposing such a system for identifying, mapping and verifying GDEs, which can be used in specific research areas with appropriate improvements. One thing to note is that we are not yet able to accurately determine the boundaries of GDEs. We can only make a comprehensive judgment based on surface vegetation and water bodies. A more precise division requires further exploration.

2. The author used data from 2020 to 2021 to divide the scope of GDEs in the Langxi River Basin. This seems to be because the author conducted the experiment during this period. Does this method also work at other times?

Answer: Thank you for your question. First of all, the distribution range of GDEs will change with the amount and range of groundwater recharge. The amount and extent of groundwater recharge will slowly change with the impacts of climate change and human activities. When natural disasters such as earthquakes, coal mining, or human activities cause serious changes in underground aquifers, the distribution range of GDEs may change significantly in a short period of time. During the period when the stratum does not change significantly, long-term groundwater level monitoring data shows that the changes in groundwater levels in the entire LRB and its vicinity are very weak. Therefore, we believe that the distribution range identification method of GDEs proposed in this article is representative. It should be noted that when the groundwater level fluctuates significantly or the stratigraphy changes significantly, we recommend re-evaluating the relevant indices and parameters in the study area and

this system.

We have added relevant discussion of this situation in the Discussion section of the paper.

In the paper:

Line 532 to 540:

The quantity and range of groundwater recharge will affect the GDEs' distribution range. The effects of climate change and human activity will gradually alter the volume and scope of groundwater recharge. The distribution range of GDEs can alter dramatically in a short amount of time when major changes are made to subterranean aquifers by natural disasters like earthquakes, coal mining, or human activity. It should be noted that when the groundwater level fluctuates significantly or the stratigraphy changes significantly, we recommend re-evaluating the relevant indices and parameters in the study area and this system. Long-term groundwater level monitoring data indicates that the changes in groundwater levels in the entire LRB and its vicinity are very weak during the period when the stratum does not change significantly. For this reason, we believe that the distribution range identification method of GDEs proposed in this article is representative.

3. The Langxi River Basin is a typical study area selected by the author. Is the GDEs identification and mapping framework proposed by the author also applicable to other river basins? I think this is also an interesting question for other readers.

Answer: We understand that readers will consider whether this approach can actually be applied to other basins. We subsequently conducted relevant experiments in another Chinese karst area, the Tanglang River Basin (TRB), and used this framework to classify GDEs. We will show you some of the results here.

Since the following content may involve some of our unpublished content, we have watermarked the images. We hope you understand.

First, we collected and organized relevant data on TRB and divided plains and mountains. Land use types, groundwater levels, water gaining and losing reaches, river bed elevation, and vegetation root systems of the TRB were investigated. The vegetation index at the end of the dry season and the wet season is used to classify crops, plants that lose water quickly, plants that lose water slowly, evergreen vegetation and buildings, etc.

[Figure]

The GDEs distribution characteristics of TRB can be further calculated by using the spatial kernel density method used in this study.

[Figure]

It should be noted that the identification, drawing and verification framework we proposed has been transplanted from LRB to TRB and has also been improved accordingly. First of all, because there are a large number of population centers and cities in TRB, we added the identification of construction land to the identification framework of GDEs. Secondly, compared with LRB, TRB has more karst caves. During the verification process of water bodies, we added work such as sampling of cave fissure water and aquatic life inside and outside caves. This can make the verification more convincing. In subsequent research, by incorporating hydrological (ecological hydrological) models that consider groundwater, and conducting source analysis of water chemistry, isotopes, and sediment, a series of research accuracy such as the identification and mapping of GDEs can be improved.

Some other formatting questions or suggestions:
1. Line 275 should add relevant introduction to the base flow segmentation method.
Answer: We have added a brief introduction to baseflow segmentation at the corresponding location in the text.
In the paper:
Base flow segmentation is a method used in hydrology to separate streamflow data into its base flow and surface runoff components. Base flow generally represents the groundwater contribution to streamflow, while surface runoff comes from precipitation events and other surface sources. There are several methods for base flow segmentation, including hydrograph separation, chemical separation, hydrometric separation, et. The study utilizes the straight-line secant method, which involves horizontally dividing the peak of the flow process line using a horizontal line. It is stipulated that the contribution of surface runoff lies above the horizontal cutting line, while the contribution of base flow lies below the horizontal cutting line. The value of the horizontal line, which represents the runoff, can be determined as the minimum flow during the dry season, the minimum daily average flow during the dry season, or the minimum monthly average flow for the year. During the non-rainy season when the karst aquifer is recharged, the flow of the spring will gradually decrease in size until it matches the recharge rate of the aquifer, as there is no additional recharge from precipitation. The equation for the flow attenuation process can be written as (Rodríguez et al., 2017).

2. It is recommended to use tables to express the data part of Line 310

Answer: Thanks to your suggestion. We have added Table 1 to sort out the types, names, resolutions, sources of data sets used in this study, and the bands used for remote sensing data.

In the paper:

Line 315:

In this paper, the data mainly includes remote sensing data, and hydrogeological survey data (Table 1).

Line 766:

Table 1: Remote sensing and hydrogeological survey data used in the research.

| Data Type | Data Name | Resolution | Resources | Bands used in research |
|---|---|---|---|---|
| **Remote sensing Data** | USGS Landsat 8 Level 2, Collection 2, Tier 1 | 30 m | Google Earth Engine (GEE) | B2-B7 (Blue, Green, Red, Near infrared, Shortwave infrared 1 and Shortwave infrared 2) |
| | NASA SRTM Digital Elevation | 30 m | GEE | Elevation and Slope (Calculated by elevation) |
| **Hydrogeological Survey Data** | Chinese stratigraphic lithology dataset | 1:2,500,000 | China Geological Survey | Geological lithology, geological body boundary, amphibole schist, crater point, et. |
| | The maximum root depth | 1:5,000,000 | Harmonized World Soil Database | - |
| | River bed level | Point scale | Field surveys | - |
| | Groundwater level | Point scale | Field surveys | - |
| | Water hydrochemical and groundwater fauna sampling | Point scale | Field surveys | - |

Note: Please see Supplementary Table 2 for remote sensing data sources.

3. Line 495 Please adjust the font size

Answer: Thank you for your suggestion. We have adjusted the font size.

4. Line 565 In the conclusion, the author uses the full names of K-GDE, S-GDE and V-GDE. In fact, the abbreviation has been used in the previous article, and it is recommended to use the abbreviation here.

Answer: Thank you for your opinion, but we have our own considerations here. As the finishing touch of a paper, the conclusion plays the role of naming the main content and summarizing the article. In this article, we reorganize the classification of GDE based on the characteristics of the study area. Using the full names of the three GDEs at the end allows readers to accept this definition (concept) more clearly.

Therefore, we believe that the full names of K-GDE, S-GDE and V-GDE can be used here.

5. The study area map of Line 710 in Figure 1 should be redesigned. It's not pretty now.
Answer: We modified Figure 1, adjusted the layout, and replaced the more difficult-to-understand geological symbols with abbreviations.
In the paper:

[Figure]

**Figure 1: The location, lithology, topography, spring water, groundwater level survey points, hydrochemical groundwater biological sampling points of the Langxi River Basin. HRAD: Holocene fluvial alluvial deposits; UPGL: Upper Pleistocene gravel layer; UCL: Upper Cambrian limestone; UCSLA: Upper Cambrian shale-limestone amalgamation.**

**Notification to the authors:**
1.Coloured or marked text in *.pdf manuscript file is not allowed. Please provide a clean version of *pdf manuscript file (with black text) with the next revision.
Answer: Thank you for your review. In the first draft, we colored citations in blue for easier reading. In accordance with journal rules, we have adjusted the text to black in the revised manuscript.

2. It seems that table is included as figure #11. If it is so, it must be re-labelled as table and the references in the manuscript text must be adjusted accordingly. A table may be inserted as an image, but still be called as a table.
Answer: As requested we have modified Figure 11 into a table, it is now Table 4.
In the paper:
**Table 4: Groundwater fauna samples. (a) *Galba.jervia.sp*; (b) *Radix lagotis Schrank*; (c) *Chironmidae*; (d)**

*Gyraulus.sp*; (e) *Dytiscidae.sp*; (f) *Anisogammarus.sp*.

| Neocaridina denticulata sinensis | | | Radix lagotis Schrank | | |
|---|---|---|---|---|---|
|  Length: 25 mm | **Phylum:** | Arthropoda |  Length: 2 mm | **Phylum:** | Mollusca |
| | **Class:** | Malacostraca | | **Class:** | Gastropoda |
| | **Order:** | Decapoda | | **Order:** | - |
| | **Family:** | Atyidae | | **Family:** | Lymnaeidae |
| | **Genus:** | *Caridina* | | **Genus:** | *Radix* |
| **Chironmidae** | | | **Gyraulus.sp** | | |
|  Length: 3 mm | **Phylum:** | Arthropoda |  Length: 1.5 mm | **Phylum:** | Mollusca |
| | **Class:** | Insecta | | **Class:** | Gastropoda |
| | **Order:** | Diptera Nematocera | | **Order:** | - |
| | **Family:** | Culicomorpha | | **Family:** | Planorbidae |
| | **Genus:** | *Chironomoidea* | | **Genus:** | *Gyraulus* |
| **Dytiscidae.sp** | | | **Anisogammarus.sp** | | |
|  Length: 0.8 mm | **Phylum:** | Arthropoda |  Length: 9 mm | **Phylum:** | Arthropoda |
| | **Class:** | Insecta | | **Class:** | Malacostraca |
| | **Order:** | Coleoptera | | **Order:** | Amphipoda |
| | **Family:** | Dytiscidae | | **Family:** | Anisogammaridae |
| | **Genus:** | *Cybister* | | **Genus:** | *Anisogammarus* |

3.Please ensure that the colour schemes used in your maps and charts allow readers with colour vision deficiencies to correctly interpret your findings. Please check your figures using the Coblis – Color Blindness Simulator (https://www.color-blindness.com/coblis-color-blindness-simulator/) and revise the colour schemes accordingly.

Answer: All images in this article have been verified using Coblis – Color Blindness Simulator. We believe that the information in the picture can be clearly identified in the three modes of Anomalous Trichromacy, Dichromatic view, and Monochromatic view. If the editorial department believes that our color matching still does not meet the regulations, please contact us and we will redraw it in the next version. Many Thanks!

4. For the next revision, please check if your figures containing photos require a copyright statement/image credit and add it to the figures (or captions) (https://publications.copernicus.org/for_authors/manuscript_preparation.html#figurest ables -> Reproduction and reuse of figures and tables). If these figures were entirely created by the authors, there is no need to add a copyright statement or credit. In that case it is important that you confirm this explicitly by email.

Answer: We have read the publication rules. We have modified one image for which reproduction rights have not been granted and certify that all images are created by us.

5. For the next revision, please make sure that information about the contribution of each of the authors of the manuscript is presented in the "Author contribution" section of the *.pdf manuscript.

Answer: As requested we have included an author contribution section in the paper.

In the paper:

**Author contribution**

Mingyang Li and Fulin Li developed the initial and final versions of this manuscript and analyzed the data. Shidong Fu, Huawei Chen, KairanWang, Xuequn Chen, and Jiwen Huang contributed their expertise and insights to oversee the analysis.

---

## Author Comment (AC4)

Responses to the reviewers

**Title:** Identification, Mapping and Eco-hydrological Signal Analysis for Groundwater-dependent Ecosystems (GDEs) in Langxi River Basin, North China (hess-2023-151)
**Authors:** Mingyang Li, Fulin Li*, Shidong Fu, Huawei Chen, Kairan Wang, Xuequn Chen, Jiwen Huang

**Thanks to the experts giving so much valuable advice, now the revision notes are shown as following.**

**RC1:**

This is a manuscript that deserves praise. It is difficult to concentrate on rich field experiments in the current setting. This study presents a framework for locating and mapping GDEs based on a mix of remote sensing, GIS, and hydrogeological field experiments in terms of research content and methodologies. The study's newly suggested normalized difference built-up and soil index, together with the difference between the wet index, may be used to assess changes in the water loss rate of plants at various stages of growth. After then, the spectrum of possible GDEs is defined by factors including slope, lithology, and elevation. Groundwater levels, river bed bottom elevation, plant root depth, karst springs, etc. were used to further infer different forms of GDEs. In the end, three factors were used to verify GDEs: groundwater biology, water chemical isotopes, and hydrological rhythm. Gaining a deeper understanding of a basin may be achieved by employing a variety of techniques to examine its biological and hydrological features. I would want to share a few small ideas and inquiries with the author.

Answer: Thank you very much for your review and the suggestions of the three experts. We have revised the paper according to each suggestion.

1. The author employed aquatic biology, hydrology, water chemistry, isotopes, and other elements to confirm. These verification findings demonstrate that spring flow and base flow in the basin have a substantial association, that the water chemistry and isotope composition of distinct water bodies vary, and that the water bodies in GDEs have comparatively separate ecosystems. It is still necessary to enhance the geographical representativeness of the aforementioned results. Is it attainable to categorize and validate GDEs in space, despite the fact that they are challenging to locate and define precisely?

Answer: Thank you very much for your comment. We will reply to your questions in the following points:

The Langxi River Basin (LRB) is a typical continuous carbonate-type karst basin in northern China. Generally speaking, this kind of basin is scattered and relatively independent. The small area contains complex topography and groundwater exchange relationships. Therefore, similar research often selects basins with a small area, which

is convenient for detailed investigation. The research methods have operational commonality in similar basins around the world. For example, this study is based on previous methods and aims at the characteristics of the underlying surface of the LRB, trying to propose a more complete research framework and improved methods. In this study, we targeted the classification and identification of GDEs according to the types of GDEs in LRB. Because different types of GDEs are distributed in LRB, including karst springs and hyporheic zones, they are more typical in northern China. In other places, affected by groundwater overexploitation and high-intensity human activities, GDEs have been seriously damaged. Even though large karst springs exist in some places, they do not form an obvious ecosystem. Therefore, first of all, we believe that choosing LRB as a typical research area is representative of the karst areas in northern China.

In this study, we propose a more systematic and comprehensive method compared to current identification and mapping research. We are not simply doing GDE classification research, because the classification is relatively simple and can be identified in the wild. And our research mainly focuses on recognition and mapping.

Based on previous research, we proposed a framework for identifying, mapping and verifying the distribution range of GDEs. This system can be improved and transplanted to other regions. For example, Barron et al. (2014) used NDVI and NDWI to classify potential GDEs in Australia. This is not significantly feasible in LRB, but this does not mean that its idea is not feasible. We selected appropriate remote sensing indices NDBSI and NDVI and jointly identified potential GDEs in LRB, and the results showed that the discrimination effect was very good. Therefore, we believe that this study is more focused on proposing such a system for identifying, mapping and verifying GDEs, which can be used in specific research areas with appropriate improvements. One thing to note is that we are not yet able to accurately determine the boundaries of GDEs. We can only make a comprehensive judgment based on surface vegetation and water bodies. A more precise division requires further exploration.

2. The author used data from 2020 to 2021 to divide the scope of GDEs in the Langxi River Basin. This seems to be because the author conducted the experiment during this period. Does this method also work at other times?

Answer: Thank you for your question. First of all, the distribution range of GDEs will change with the amount and range of groundwater recharge. The amount and extent of groundwater recharge will slowly change with the impacts of climate change and human activities. When natural disasters such as earthquakes, coal mining, or human activities cause serious changes in underground aquifers, the distribution range of GDEs may change significantly in a short period of time. During the period when the stratum does not change significantly, long-term groundwater level monitoring data shows that the changes in groundwater levels in the entire LRB and its vicinity are very weak. Therefore, we believe that the distribution range identification method of GDEs proposed in this article is representative. It should be noted that when the groundwater level fluctuates significantly or the stratigraphy changes significantly, we

recommend re-evaluating the relevant indices and parameters in the study area and this system.

We have added relevant discussion of this situation in the Discussion section of the paper.

**In the revised paper:**

Page 18 Line 564 to 572:

The quantity and range of groundwater recharge will affect the GDEs' distribution range. The effects of climate change and human activity will gradually alter the volume and scope of groundwater recharge. The distribution range of GDEs can alter dramatically in a short amount of time when major changes are made to subterranean aquifers by natural disasters like earthquakes, coal mining, or human activity. It should be noted that when the groundwater level fluctuates significantly or the stratigraphy changes significantly, we recommend re-evaluating the relevant indices and parameters in the study area and this system. Long-term groundwater level monitoring data indicates that the changes in groundwater levels in the entire LRB and its vicinity are very weak during the period when the stratum does not change significantly. Therefore, the Four diagnostic criteria framework proposed in the study can effectively identify GDEs in areas such as grasslands, deserts, plains, and karsts in arid, semi-arid, and sub-humid areas.

3. The Langxi River Basin is a typical study area selected by the author. Is the GDEs identification and mapping framework proposed by the author also applicable to other river basins? I think this is also an interesting question for other readers.

Answer: We understand that readers will consider whether this approach can actually be applied to other basins. We subsequently conducted relevant experiments in another Chinese karst area, the Tanglang River Basin (TRB), and used this framework to classify GDEs. We will show you some of the results here.

Since the following content may involve some of our unpublished content, we have watermarked the images. We hope you understand.

First, we collected and organized relevant data on TRB and divided plains and mountains. Land use types, groundwater levels, water gaining and losing reaches, river bed elevation, and vegetation root systems of the TRB were investigated. The vegetation index at the end of the dry season and the wet season is used to classify crops, plants that lose water quickly, plants that lose water slowly, evergreen vegetation and buildings, etc.

[Figure]

The GDEs distribution characteristics of TRB can be further calculated by using the spatial kernel density method used in this study.

[Figure]

It should be noted that the identification, drawing and verification framework we proposed has been transplanted from LRB to TRB and has also been improved accordingly. First of all, because there are a large number of population centers and cities in TRB, we added the identification of construction land to the identification framework of GDEs. Secondly, compared with LRB, TRB has more karst caves. During the verification process of water bodies, we added work such as sampling of cave fissure water and aquatic life inside and outside caves. This can make the verification more convincing. In subsequent research, by incorporating hydrological (ecological hydrological) models that consider groundwater, and conducting source analysis of water chemistry, isotopes, and sediment, a series of research accuracy such as the identification and mapping of GDEs can be improved.

Some other formatting questions or suggestions:
1. Line 275 should add relevant introduction to the base flow segmentation method.
Answer: We have added a brief introduction to baseflow segmentation at the corresponding location in the text.
**In the revised paper:**
Page 9 Line 267 to 270:

Base flow segmentation is a method used in hydrology to separate streamflow data into its base flow and surface runoff components. Base flow generally represents the groundwater contribution to streamflow, while surface runoff comes from precipitation events and other surface sources. There are several methods for base flow segmentation, including hydrograph separation, chemical separation,

hydrometric separation, et. The study utilizes the straight-line secant method, which involves horizontally dividing the peak of the flow process line using a horizontal line. It is stipulated that the contribution of surface runoff lies above the horizontal cutting line, while the contribution of base flow lies below the horizontal cutting line. The value of the horizontal line, which represents the runoff, can be determined as the minimum flow during the dry season, the minimum daily average flow during the dry season, or the minimum monthly average flow for the year. During the non-rainy season when the karst aquifer is recharged, the flow of the spring will gradually decrease in size until it matches the recharge rate of the aquifer, as there is no additional recharge from precipitation. The equation for the flow attenuation process can be written as (Rodríguez et al., 2017).

2. It is recommended to use tables to express the data part of Line 310
Answer: Thanks to your suggestion. We have added Table 1 to sort out the types, names, resolutions, sources of data sets used in this study, and the bands used for remote sensing data.

**In the revised paper:**
Page 11 Line 317:
In this paper, the data mainly includes remote sensing data, and hydrogeological survey data (Table 1).

Page 33 Line 810:
**Table 1: Remote sensing and hydrogeological survey data used in the research.**

| Data Type | Data Name | Resolution | Resources | Bands used in research |
|---|---|---|---|---|
| **Remote sensing Data** | USGS Landsat 8 Level 2, Collection 2, Tier 1 | 30 m | Google Earth Engine (GEE) | B2-B7 (Blue, Green, Red, Near infrared, Shortwave infrared 1 and Shortwave infrared 2) |
| | NASA SRTM Digital Elevation | 30 m | GEE | Elevation and Slope (Calculated by elevation) |
| **Hydrogeological Survey Data** | Chinese stratigraphic lithology dataset | 1:2,500,000 | China Geological Survey | Geological lithology, geological body boundary, amphibole schist, crater point, et. |
| | The maximum root depth | 1:5,000,000 | Harmonized World Soil Database | - |
| | River bed level | Point scale | Field surveys | - |
| | Groundwater level | Point scale | Field surveys | - |
| | Water hydrochemical and groundwater fauna sampling | Point scale | Field surveys | - |

Note: Please see Supplementary Table 2 for remote sensing data sources.

3. Line 495 Please adjust the font size

Answer: Thank you for your suggestion. We have adjusted the font size.

4. Line 565 In the conclusion, the author uses the full names of K-GDE, S-GDE and V-GDE. In fact, the abbreviation has been used in the previous article, and it is recommended to use the abbreviation here.

Answer: Thank you for your opinion, but we have our own considerations here. As the finishing touch of a paper, the conclusion plays the role of naming the main content and summarizing the article. In this article, we reorganize the classification of GDE based on the characteristics of the study area. Using the full names of the three GDEs at the end allows readers to accept this definition (concept) more clearly. Therefore, we believe that the full names of K-GDE, S-GDE and V-GDE can be used here.

5. The study area map of Line 710 in Figure 1 should be redesigned. It's not pretty now.

Answer: We modified Figure 1, adjusted the layout, and replaced the more difficult-to-understand geological symbols with abbreviations.

**In the revised paper:**

[Figure]

**Figure 1: The location, lithology, topography, spring water, groundwater level survey points, hydrochemical groundwater biological sampling points of the Langxi River Basin. HRAD: Holocene fluvial alluvial deposits; UPGL: Upper Pleistocene gravel layer; UCL: Upper Cambrian limestone; UCSLA: Upper Cambrian shale-limestone amalgamation.**

**RC2:**

The manuscript by Li et al. focuses on the identification and analysis of groundwater-dependent ecosystems (GDEs) in a specific region. The authors propose a four-diagnostic criteria framework for identifying GDEs based on remote sensing, GIS data dredging, and hydrogeological surveys. Overall, I think this work will be of interest to the HESS readership but needs some reworking to be considered further for publication. I make a few suggestions for improvement below:

Answer: Thank you very much for your review and the suggestions. We have revised the paper according to each suggestion. In response to your opinions, we have made the following point-to-point modifications, which can be summarized as:

(1) We have adjusted the arrangement of the conclusion section to make the research findings more like an article rather than a report listing the research results. The three chapters are now: 3.1 Hydrogeological investigation of three types of GDEs; 3.2 Distribution and mapping of GDEs; 3.3 Ecohydrological signals of GDEs.

(2) We have strengthened the connection with the method in the results section, and related the formulas in the method to the results. On the basis of adjusting the order of the results part, the description in the results was significantly modified to make the entire article more fluent.

(3) We have redrawn Figure 2, Figure 8 and Figure 10, adding a lot of details. This makes our images and texts more relevant.

(4) In the discussion section, we added a comparative analysis with two existing similar studies, analyzed the advantages and disadvantages of this study and existing studies, and further illustrated the advancement of the Four diagnostic criteria framework.

1. The result section lacks clarity and structure. It resembles a report rather than a concise summary of findings. Some materials in the results, particularly those pertaining to the authors' methodologies or rationales, such as those indicated on Page 11 Lines 337-339, Page 14 Lines 407-408, and Page 14 Lines 421-423, should be relocated to the method section to enhance organization and coherence.

Answer: Thank you for your opinion. To improve the clarity of the results section of the article, we modified revised the arrangement of chapters. The results section of this article now has five chapters. Please allow us to introduce the structural ideas of the results section of this article. The results part of this article is arranged according to the framework of the research method in Figure 2. First, Section 3.1 introduces the hydrogeological investigation results for three types of GDEs. Section 3.2 introduces the distribution and mapping of GDEs. Section 3.3 introduces the ecohydrological signals of GDEs to verify whether the GDEs distribution is correct.

Then, we have rewritten the results section in relation to the methods and rationale of Chapter 2 to enhance organization and coherence.

(1) Page 11 Lines 337-339 of the **original text**

As the first sentence of Section 3.2, in order to better connect the methods and results,

it should also serve as the beginning to guide the rest of the text. We've rewritten this.

**In the revised paper:**

Page 13 Line 379-386

Based on the research framework in Figure 3, the study first uses remote sensing indicators such as terrain, NDVI, NDWI, and DWN to identify characteristics such as waters, bare land, wetlands, and vegetation to determine the potential distribution of GDE. Supplementary Figure 2 displays the distribution characteristics of NDVI and NDWI in the study area at the end of the dry and wet seasons. In the central and southern plains of the study area, NDVI remained high at the end of the wet season (Supplementary Figure 2a) and slightly decreased at the end of the dry season (Supplementary Figure 2b), indicating that the vegetation in this area primarily experiences rapid drying.

(2) Page 14 Lines 407-408 of the **original text**

We did not clearly express what we wanted to describe in the original text. We've rewritten this.

**In the revised paper:**

Page 12 Line 345-348

The research acquired information about the maximum vegetation root depth, river bottom elevation, and groundwater level of the LRB (Figure 5a) by the gathering of GIS data and basin hydrogeological survey. We qualitatively evaluated the gaining and losing river portions using data on the elevation of the river bed and the depth of the groundwater table (Figure 5b).

The analysis of underground water table depths reveals that the shallow water table area (0 to 5m) is primarily located in the middle of the basin where the tributaries converge, and numerous karst springs are situated nearby. The vegetation in LRB is predominantly composed of deciduous broad-leaved forest and deciduous open shrubs, with relatively developed underground root systems owing to the year-round flow of rivers. The maximum root depth in the basin ranges from 1.5 to 2.5 m, with some exceeding 4.5 m. Areas with deeper root depths are present on both sides of the river. In the lower reaches of the Langxi River near the Yellow River, the roots of vegetation are relatively shallow, consistent with the spatial distribution of surface lithology. However, the reason for the shallow roots is not caused by the surface lithology. In fact, it is due to the abundance of water in the area, and a large number of farmlands are distributed here. The riverbed bottom level changes more gently from upstream to downstream compared to the change in elevation. Based on the identification of potential GDEs, the study was able to accurately divide S-GDE and V-GDE.

(3) Page 14 Lines 421-423 of the **original text**

This paragraph in the original article is not closely related to the previous article, so we have rewritten this paragraph.

**In the revised paper:**

Page 14 Line 431-437

Based on the previous ecological and hydrogeological survey results, the study subdivided potential GDEs into S-GDEs, V-GDEs and K-GDEs according to the four diagnostic criteria framework (Figure 3), and used the spatial kernel density function for mapping. The distribution results of GDEs in the watershed are shown in Figure 8. Green, orange and magenta represent V-GDEs, K-GDEs and S-GDEs respectively. The depth of the color represents the spatial core density of the GDE. When the climate is dry, areas with lower spatial core density will gradually no longer belong to the scope of the GDE. The GDEs in the basin are mainly located in the central and western parts of the LRB, covering an area of approximately 49 km$^2$, which accounts for 29% of the basin's total area.

(4) We have also made corresponding modifications to some sentences that were not mentioned by experts in this article.

**In the revised paper:**

Page 1 Line 23-33

In order to verify the reliability of GDE distribution, the study verified the determination of GDEs through hydrological rhythm analysis, hydrochemical characteristics analysis of various water bodies in the basin, and ecohydrological signals such as groundwater invertebrates. The hydrological rhythm analysis in Shuyuan section showed that the proportion of base flow to river flow is about 54.15% and S-GDEs still receive spring water recharge even in the extremely dry season. And the analysis of hydrochemical sampling from the karst aquifer, Quaternary aquifer, spring water and surface reservoir water reveals that GDEs are also relished by groundwater. More important, we also found a distinctive ecohydrological signal of GDEs is the presence of millimeter-sized groundwater fauna living in the different types of GDEs. In addition, the study believes that the use of isotope and environmental DNA technology to analyze the hydrological-sediment-biological connectivity between groundwater and GDE is the future development direction of this field.

2. The current version lacks substantial supporting information on ecohydrological signals aiding in mapping GDEs. Section 3.4 primarily focuses on the spatial distribution of GDEs within the study area. Sections 3.5 and 3.6 delve into detailed descriptions of hydro-biogeochemical features observed in GDEs. However, there appears to be a weak connection between these descriptions and the verification of various GDEs, as depicted in Figure 2. Strengthening this connection is essential for a more cohesive presentation of the study's findings regarding the role of ecohydrological signals in GDE mapping.

Answer: Thank you for your opinion, your opinion is very important to our modifications. We recognized the shortcomings of the original version of the weak connection of ecohydrological signals in GDEs drawing and made the following modifications:

(1) The section shown in Figure 2 has important connections with Section 3.3.1. Section 3.3.1 describes the lateral recharge relationship of groundwater to the Langxi River. In response to this, this study conducted a geological survey on the Shuyuan Spring and Langxi River sections, and explored the relationship between the groundwater level and Shuyuan Spring, as well as relevant information about the stratigraphy of the sections. In the text, we reformulate the connection between Figure 2 and Section 3.3.1. And in order to increase relevance, we modified Figure 2 to more clearly represent the three types of GDEs present in the modified typical section.

**In the revised paper:**
Page 15 Line 451-456

The Shuyuan Spring and Langxi River profile (hereinafter referred to as the Shuyuan Spring profile, Figure 2) shows the location and elevation relationship between groundwater and the Langxi River profile. The eastern Cambrian Zhushadong-Zhangxia Formation limestone is rich in groundwater buried deeper than the river. Shuyuan Spring, one of the descending springs in the basin, gushes out at the intersection of the Quaternary sedimentary layer and the Cambrian Zhushadong-Zhangxia Formation limestone. At the same time, groundwater also recharges to the Langxi River along the stratigraphic fissures.

**In the revised paper:**
Figure 2.

[Figure]

Figure 2: Hydrogeological profile of Shuyuan spring in LRB. The dotted line shows the characteristics of the water table in the geological section. The geological types in the figure are Q: Quaternary sedimentary layer; ЄjZ: Cambrian Zhushadong-Zhangxia Formation limestone; ЄcM: Cambrian Gushan-Chaomidian Formation limestone.

(2) Sections 3.3.2 and 3.3.3 respectively use the hydrochemical characteristics of different water bodies and the groundwater fauna to verify the identification of GDEs. In the original text, we did not reflect the corresponding connection well.

In the revised manuscript, we first modified Figure 10 in Section 3.3.2. We classified the water chemical characteristics corresponding to the distribution of the three GDEs in Figure 8 and discussed them accordingly in the text.

**In the revised paper:**
Page 16 Line 495-504:

Combined with the distribution range of the three GDEs and the water sample collection locations in Figure 8, we can clearly see that the hydrochemical characteristics of water bodies of the same GDE type have obvious clustering relationships. The overlapping relationship of hydrochemical characteristics in GDE groups is consistent with the distribution characteristics of GDEs types at the spatial scale of the basin (Figure 10a), which is also reflected in the cluster analysis diagram (Figure 10b). In addition, descending springs like Shuyuan Spring are not only closely related to karst aquifers, but also have good hydraulic connections with Quaternary aquifers. On the other hand, Huiquan Reservoir is a surface reservoir stored in a river barrage and receives a large amount of groundwater recharge. Therefore, it exhibits hydrochemical characteristics similar to spring water and Quaternary pore water. It can be seen that the interaction between surface water and groundwater in this area is strong, and there are obvious differences in the hydrochemical characteristics of GDE, but there are also certain similarities.

**In the revised paper:**
Page 30 Line 795:

[Figure]

Figure 8: Langxi River Basin GDEs distribution area and ecohydrological signals.

**In the revised paper:**
Page 32 Line 805:

[Figure]

Figure 10: Hydrochemical characteristics of ten water samples in LRB.

**In the revised paper:**
Page 17 Line 510-516:

Finding groundwater fauna near karst caves is easy because there is an abundance of food, making it easier for them to survive. Hu Spring, a natural karst cave, is home to three species of Chironomid larvae; Anisogammarus sp.; Radix lagotis were found in it. Typically, these faunas mainly comprise of Arthropods, Coelenterates and Mollusks that live in groundwater throughout their entire life cycle, and are known as stygobites, a true groundwater fauna. Combining the spatial distribution range of GDEs in Figure 8 of this study and the groundwater fauna in Tables 3 and 4, it can be seen that these groundwater fauna not only appear in the range of K-GDEs with caves and karst springs, but also in the range of some S-GDEs and a small amount of V-GDEs.

3. The proposed framework should be compared to other existing approaches to highlight its potential advantages and limitations. This discussion is crucial for providing insights into the novelty and effectiveness of the proposed methodology.

Answer: Thank you for your opinion. In the original article, we discussed the differences between this study and Barron et al. (2014)'s study, and explored the applicability of NDVI and NDWI in the identification of GDEs. In response to the experts' opinions, we conducted a comparative analysis with existing similar research from the perspective of input variables of the identification model and verification of GDEs. In the discussion section, it was proposed that this study can continue to supplement input to improve the accuracy of GDEs identification, and the superiority of this study in using hydrological rhythms, hydrochemical characteristics, and groundwater fauna for GDEs verification.

**In the revised paper:**
Page 18 Line 544-555:
Compared with the study of El-Hokayem et al. (2023) on identifying GDEs in Italy, this study did not include climate factors into the identification indicators. Instead, it focused more on elements of the local watershed itself, such as water body characteristics and groundwater biological characteristics. This is because the study area of El-Hokayem et al. (2023) belongs to the coastal area, and the vegetation responds more sensitively to climate elements. Our research prefers to identify and establish direct connections between groundwater-surface water-surface/near-surface ecosystems through hydrochemical characteristics, and then verify the identification scope of GDEs.

The research results of Duran-Llacer et al. (2022) using more than ten terrain and remote sensing indicators to identify and map GDEs in Chile showed that increasing the input variables of the identification model can help improve the accuracy of identification. This can be used as a reference for the application of the identification framework proposed in this study in other basins. The multi-index verification system of hydrological rhythm, water chemistry, and groundwater fauna used in this study can verify the accuracy of GDEs identification from multiple dimensions, which has basically not been used in existing research. And this will help provide stronger support for the results for verification in future relevant studies.

Specific comments Page 4, Line 100: Delete ", etc".
Answer: Here, there are two etc. in the original text, which may cause ambiguity to readers. We have corrected the inappropriate expression.

**In the revised paper:**
Page 3-4 Line 97-100:
These characteristics can often be regarded as a specific signal for monitoring ecosystem status and linking the functions of organisms to ecohydrological processes, such as the rhythm of hydrometeorological elements, hydrogeochemical

characteristics, and biological indicators (just like biodiversity, connectivity), etc.

Page 6, Line 183: Please clarify the scale behind select a 10-degree angle.

Answer: In the original article, we briefly stated "The determination of this parameter can be manually adjusted based on one-third of the average slope of the basin until the plains and mountains are clearly distinguished." In response to experts' suggestions, we first put the method of parameter determination and the results of selecting different parameters into a supplementary file. This will allow readers to better understand my approach.

**In the revised paper:**
Page 6 Line 185-187:

where, $grid_{plain}$ represents the grid divided into plains; $\Delta_{slope}$ is the threshold of

the maximum plain slope, in this paper we take $\Delta_{slope} = 10°$. The determination of

this parameter can be manually adjusted based on one-third of the average slope of the basin until the plains and mountains are clearly distinguished (See Supplementary material I).

**In the Supplementary material I:**
Below, Figure S1 shows our results of plain and hill classification using 4 different slopes. It can be seen that except for minor differences in details, overall slope does not have a great impact on plains and hills in LRB. This is due to the fact that the mean elevation limits the role of slope in this basin.

[Figure]

(a) $\Delta_{slope} = 5°$    (b) $\Delta_{slope} = 10°$    (c) $\Delta_{slope} = 15°$    (d) $\Delta_{slope} = 20°$

Figure S1. Different slopes distinguish the plain and hilly results of LRB.

Page 10, Line 285: Delete "2"

Answer: Thanks for your opinion, but the 2 here is not a typo. We have modified the way this sentence is expressed. The original text means that we use HNO3 to adjust the pH value of the solution to 2.

**In the revised paper:**

Page 10 Line 291-292:

Subsequently, all the water samples were passed through a 0.45 μm filter, and the liquid samples were acidified the pH to 2 using pure HNO3 to prevent the precipitation of metals before metal analysis.

Page 16, Line 495: Font size is not consistent.

Answer: Thank you for your suggestion. We have adjusted the font size.

**RC3:**

I have had the opportunity to review the manuscript detailing the assessment of Groundwater Dependent Ecosystems (GDEs) using remote sensing data and hydrogeological surveys in the Langxi River Basin. This study provides an insightful exploration of the methods used for identifying and characterizing GDEs, a topic of great environmental importance due to the critical role these ecosystems play in biodiversity conservation and water resource management. The findings are robust and contribute meaningfully to the field, especially in terms of methodological innovations. However, I have several comments that I believe could further refine and enhance the value of the work presented.

**Answer:** Thank you very much for your review of our manuscript and your valuable comments. Below we will make modifications one by one based on your comments.

1. The manuscript offers a detailed account of the combined use of GIS and hydrogeological data. However, it would benefit from a more explicit explanation of the choice and configuration of the remote sensing technologies and the rationale behind the specific indices selected for analysis. Providing this information would help in replicating the study in other regions or contexts, enhancing its utility for broader application.

**Answer:** Thank you very much for your comments. We have made the following changes to this content. (1) A table was added to the data section to describe the datasets used and the properties of each dataset.

**In the revised paper:**

Page 11 Line 317:

In this paper, the data mainly includes remote sensing data, and hydrogeological survey data (Table 1).

Page 33 Line 810:

**Table 1: Remote sensing and hydrogeological survey data used in the research.**

| Data Type | Data Name | Resolution | Resources | Bands used in research |
|---|---|---|---|---|
| **Remote sensing Data** | USGS Landsat 8 Level 2, Collection 2, Tier 1 | 30 m | Google Earth Engine (GEE) | B2-B7 (Blue, Green, Red, Near infrared, Shortwave infrared 1 and Shortwave infrared 2) |
| | NASA SRTM | 30 m | GEE | Elevation and Slope |

| | Digital Elevation | | | (Calculated by elevation) |
|---|---|---|---|---|
| **Hydrogeological Survey Data** | Chinese stratigraphic lithology dataset | 1:2,500,000 | China Geological Survey | Geological lithology, geological body boundary, amphibole schist, crater point, et. |
| | The maximum root depth | 1:5,000,000 | Harmonized World Soil Database | - |
| | River bed level | Point scale | Field surveys | - |
| | Groundwater level | Point scale | Field surveys | - |
| | Water hydrochemical and groundwater fauna sampling | Point scale | Field surveys | - |

Note: Please see Supplementary Table 2 for remote sensing data sources.

(2) Additional explanations were provided for the reasons for using the relevant methods and content locations.

**In the revised paper:**
Page 6 Line 180:
Groundwater collection places are typically found on level plains or low-lying valleys. To make these regions easier to locate, lowlands and mountains have been divided. First, using digital elevation model (DEM) and slope (calculated by DEM), we can distinguish the plains and hills of the basin (Eq.1), and further divide the plains of shallow fissure rocks according to the surface lithology, which is the area with the conditions for the formation of GDEs.

(3) We have added a brief introduction to baseflow segmentation at the corresponding location in the text.
**In the revised paper:**
Page 9 Line 267 to 270:
Base flow segmentation is a method used in hydrology to separate streamflow data into its base flow and surface runoff components. Base flow generally represents the groundwater contribution to streamflow, while surface runoff comes from precipitation events and other surface sources. There are several methods for base flow segmentation, including hydrograph separation, chemical separation, hydrometric separation, et. The study utilizes the straight-line secant method, which involves horizontally dividing the peak of the flow process line using a horizontal line. It is stipulated that the contribution of surface runoff lies above the horizontal cutting line, while the contribution of base flow lies below the horizontal cutting line.

(4) Additionally, we have made certain changes to the introduction of pertinent techniques, etc., based on the feedback from RC1 and RC2. Owing to limited space, we won't reiterate them here.

2. The results section effectively illustrates the application of NDWI and NDVI indices in identifying GDEs. However, ensuring that all figures and tables consistently reflect the descriptions provided in the text would improve the manuscript's readability and professional appearance. Specifically, ensuring that legends and captions are clear and that graphical representations directly correspond to the described findings is crucial.

**Answer:** In this paper, Supplementary Figure 2 shows the values of NDVI and NDWI at the end of the dry and wet seasons in the Langxi River Basin (LRB). Figure 6 (subject to the revised manuscript number) shows the change rates of WET and NDBSI at the end of the dry and wet seasons. (1) These two figures have their own necessity. These two figures can illustrate the changes in vegetation moisture in the LRB at the end of the dry and wet seasons. (2) In terms of numerical values, the extreme values of the WET and NDBSI indices may have made you suspicious. After checking, the numerical values in Figure 6 and Supplementary Figure 2 are correct. It can be seen that the area with the largest change in WET at the end of the dry and wet season is in the plains of the basin close to the ridges, while the area with the largest change in NDBSI is basically in the ridge area. This illustrates two points. First, in these high-lying areas, vegetation cannot be replenished by groundwater and loses water rapidly at the end of the growing season. Second, it indirectly proves the correctness of the first step of the GDEs diagnostic framework proposed in this study to divide plains and mountains. (3) Legend and title. After checking, the content and location of the legend in the figure are correct. This is the first time that NDBSI appears in the title, so we have added its full name in the title of Figure 6. If you feel that we have not fully understood the meaning of your comment, please contact us.

3. While the study discusses the settings used within the numerical models and their boundary conditions, linking these directly to either field conditions or the parameters set during laboratory tests could strengthen the study. A detailed comparison would not only validate the model further but also provide clarity on its applicability in real-world scenarios.

**Answer:** Thank you for your constructive comments. In the discussion section, combined with the opinions of the other two experts, we first compared similar studies, pointed out the differences and innovations of this study and other studies, and also analyzed the shortcomings of this study. We also analyzed the effects of different values of the parameters in the model (Eq. 1) (Supplementary material I). Based on your comments, we further discussed the connection between the model and indoor experiments.

**In the revised paper:**
Page 19 Line 581 to 583:
Given that identifying GDEs is a fundamental issue, some authors have also examined the impact of groundwater extraction and other factors (Gou et al., 2015; Münch and Conrad, 2007; Pérez Hoyos et al., 2016). We think that future studies can

quantitatively link groundwater and ecosystem through the use of stable hydrogen and oxygen isotope experimental data on vegetation's absorption of groundwater in conjunction with ecological or ecohydrological models.

4. The conclusion successfully outlines the study's cont ributions to understanding GDEs. However, integrating specific recommendations for policymakers on managing groundwater extraction in coastal areas could significantly enhance the manuscript's impact. Practical guidelines based on the study's findings would be invaluable for regional planning and conservation efforts.

**Answer:** Thank you very much for your suggestion. By studying the spatial distribution characteristics of GDEs, this paper can determine the contribution of groundwater to the ecosystem at a spatial scale. The research results have a guiding role for decision makers in groundwater exploitation in karst areas. We briefly supplemented the significance of this study in this regard in the paper. We believe that in subsequent research, our research can truly make guiding contributions to government and other functional departments.

**In the revised paper:**
Page 20 Line 623 to 625:
However, the knowledge gap regarding the ecohydrological connectivity between groundwater and GDE can be improved by utilizing isotope analysis, stygofauna tracing, and DNA sequencing technology under the recommended four-diagnostic criteria framework in the future. The research helps to better understand the connection between groundwater and ecosystem, and the results can guide decision makers in groundwater exploitation in karst areas.

5. The manuscript exhibits a commendable level of detail and scientific rigor. However, to elevate the manuscript's clarity and professional presentation, a comprehensive review of typographical errors, figure consistency, and data presentation is essential. For example, some figures and diagrams exhibit inconsistent use of color schemes and font sizes, which could potentially confuse the reader or detract from the data's visual impact. Additionally, inconsistencies in symbol usage and abbreviation definitions across the text and figures were noted. Ensuring that all graphical representations adhere to a uniform style guide would significantly enhance readability and the visual appeal of the manuscript. It is also recommended to verify the accuracy of all legends and captions to ensure they precisely describe the corresponding figures and tables. Addressing these issues will not only refine the presentation but also bolster the manuscript's overall credibility and ease of understanding.

**Answer:** Thank you very much for your recognition. In response to this comment, we have made the following changes. (1) We have modified the sentences with inconsistent fonts. (2) We have made a lot of changes to the figures in the article, including unifying the color usage of the three types of GDEs (i.e., yellow for K-GDE, green for V-GDE, and magenta for S-GDE). We have modified the presentation of

Figure 2 and used a schematic diagram of the three types of GDEs to highlight the importance of the typical sections of the basin. We have added the judgment conditions of spring types and GDEs in Figure 8. The three types of GDEs are circled in Figure 10.

In summary, this manuscript provides important insights into the assessment and management of GDEs using advanced remote sensing and hydrogeological methods. Addressing the points listed above could strengthen the manuscript's impact, making it a significant contribution to environmental science research. I look forward to seeing these enhancements in the revised manuscript.

**Answer:** Thank you very much for your high evaluation of our manuscript. We have made point-by-point revisions based on the opinions of the three experts. If there are any inadequate revisions, please communicate with us in a timely manner.

**Notification to the authors:**

1.Coloured or marked text in *.pdf manuscript file is not allowed. Please provide a clean version of *pdf manuscript file (with black text) with the next revision.

Answer: Thank you for your review. In the first draft, we colored citations in blue for easier reading. In accordance with journal rules, we have adjusted the text to black in the revised manuscript.

2. It seems that table is included as figure #11. If it is so, it must be re-labelled as table and the references in the manuscript text must be adjusted accordingly. A table may be inserted as an image, but still be called as a table.

Answer: As requested we have modified Figure 11 into a table, it is now Table 4.

**In the revised paper:**

**Page 35 Line 814:**

**Table 4: Groundwater fauna samples. (a) *Galba.jervia.sp*; (b) *Radix lagotis Schrank*; (c) *Chironmidae*; (d) *Gyraulus.sp*; (e) *Dytiscidae.sp*; (f) *Anisogammarus.sp*.**

| Neocaridina denticulata sinensis | | | Radix lagotis Schrank | | |
|---|---|---|---|---|---|
| (a) Length: 25 mm | **Phylum:** | Arthropoda | (b) Length: 2 mm | **Phylum:** | Mollusca |
| | **Class:** | Malacostraca | | **Class:** | Gastropoda |
| | **Order:** | Decapoda | | **Order:** | - |
| | **Family:** | Atyidae | | **Family:** | Lymnaeidae |
| | **Genus:** | *Caridina* | | **Genus:** | *Radix* |
| Chironmidae | | | Gyraulus.sp | | |
| (c) Length: 3 mm | **Phylum:** | Arthropoda | (d) Length: 1.5 mm | **Phylum:** | Mollusca |
| | **Class:** | Insecta | | **Class:** | Gastropoda |
| | **Order:** | Diptera Nematocera | | **Order:** | - |
| | **Family:** | Culicomorpha | | **Family:** | Planorbidae |
| | **Genus:** | *Chironomoidea* | | **Genus:** | *Gyraulus* |

| Dytiscidae.sp | | | Anisogammarus.sp | | |
|---|---|---|---|---|---|
| (e) Length: 0.8 mm | **Phylum:** | Arthropoda | (f) Length: 9 mm | **Phylum:** | Arthropoda |
| | **Class:** | Insecta | | **Class:** | Malacostraca |
| | **Order:** | Coleoptera | | **Order:** | Amphipoda |
| | **Family:** | Dytiscidae | | **Family:** | Anisogammaridae |
| | **Genus:** | *Cybister* | | **Genus:** | *Anisogammarus* |

3.Please ensure that the colour schemes used in your maps and charts allow readers with colour vision deficiencies to correctly interpret your findings. Please check your figures using the Coblis – Color Blindness Simulator (https://www.color-blindness.com/coblis-color-blindness-simulator/) and revise the colour schemes accordingly.

Answer: All images in this article have been verified using Coblis – Color Blindness Simulator. We believe that the information in the picture can be clearly identified in the three modes of Anomalous Trichromacy, Dichromatic view, and Monochromatic view. If the editorial department believes that our color matching still does not meet the regulations, please contact us and we will redraw it in the next version. Many Thanks!

4. For the next revision, please check if your figures containing photos require a copyright statement/image credit and add it to the figures (or captions) (https://publications.copernicus.org/for_authors/manuscript_preparation.html#figurest ables -> Reproduction and reuse of figures and tables). If these figures were entirely created by the authors, there is no need to add a copyright statement or credit. In that case it is important that you confirm this explicitly by email.

Answer: We have read the publication rules. We have modified one image for which reproduction rights have not been granted and certify that all images are created by us.

5. For the next revision, please make sure that information about the contribution of each of the authors of the manuscript is presented in the "Author contribution" section of the *.pdf manuscript.

Answer: As requested we have included an author contribution section in the paper.
**In the revised paper:**
**Page 21 Line 634 to 636:**
Author contribution

Mingyang Li and Fulin Li developed the initial and final versions of this manuscript and analyzed the data. Shidong Fu, Huawei Chen, KairanWang, Xuequn Chen, and Jiwen Huang contributed their expertise and insights to oversee the analysis.

---

## Author Response (AR2)

**Responses to the reviewers**

**Title:** Identification, Mapping and Eco-hydrological Signal Analysis for Groundwater-dependent Ecosystems (GDEs) in Langxi River Basin, North China (hess-2023-151)

**Authors:** Mingyang Li, Fulin Li*, Shidong Fu, Huawei Chen, Kairan Wang, Xuequn Chen, Jiwen Huang

**Thanks to the experts giving so much valuable advice, now the revision notes are shown as following.**

**Public justification (visible to the public if the article is accepted and published):**

Dear authors, thanks for the revised version. The referee was happy with the improvements. I agree. So for me the paper is almost ready for publication. I have two technical issues I would like you to look at. In Figure 9 c, you give three recession line with R2. The first one (red color) is a line between 2 points, with R2=1. I hope you agree with me this is not allowed and not informative. So I suggest to delete this first recession equation and R2. Second, all figure have quite strong colors, please check with Copernicus if these color schemes are acceptable for Hess (colorblind color schemes are asked for). Especially fig 3, with colors fading and text in color could be improved.

I also suggest to delete the words 'min' and 'max' along several of the axis in figure 7.

Legends in Fig 2 and 7: please take legend/text out of the figure and into the caption or put the legend below the figure (lithology explanation for example). In Figure 7, please write the legend in the caption, not in color in the figure. This makes the figure very hard to read.

Kind regards

Thom Bogaard

Answer: Thank you very much for your suggestions. We have revised these four figures according to each suggestion.

1.For Figure 2, we moved the legend out of the image, placed it on top, and simplified the text.

For Figure 3, we changed the text to a uniform black color that is easy to identify, and deleted the colors that affect the resolution in some frames.

For Figure 7, we deleted a lot of text in the image and the "max" and "min" on the coordinate axis, and added a legend above the image.

For Figure 9, we deleted the straight line equation with only two points and replaced the original correlation coefficient R with $R^2$.

[Figure]

**Figure 2: Hydrogeological profile of Shuyuan spring in LRB. The dotted line shows the characteristics of the water table in the geological section. The geological types in the figure are Q: Quaternary sedimentary layer; ЄjZ: Cambrian Zhushadong-Zhangxia Formation limestone; ЄcM: Cambrian Gushan-Chaomidian Formation limestone.**

[Figure]

**Figure 3: Diagnostic framework for GDEs identification, mapping and verifying. Note: DEM: digital elevation model; DWN: the difference between wet index and the normalized difference built-up and soil index; GIS: geographic information system; SL: surface lithology; RB: river bed; GW: groundwater; WHS: water hydrochemical sampling; GFS: groundwater fauna sampling; WL & WQ: water level and water quality; S-GDEs: stream-type GDEs; V-GDEs: vegetation-type GDEs; K-GDEs: karst-type GDEs.**

[Figure]

**Figure 7: The centroid scatterplots for the difference between WET and NDBSI (a), NDVI (b) and NDWI (c) in the end of the wet season and the dry season (2020 to 2021).**

[Figure]

**Figure 9: (a) Relationship between precipitation and Shuyuan section river flow and its base flow from 1990 to 2015; (b) Hydrologic graph of Shuyuan spring from July 1993 to July 1994; (c) Spring discharge attenuation curve of Shuyuan spring from November 1993 to April 1994.**

2. The images in this article use a variety of colors. Below is the performance of each image in Color Blindness Simulator. We believe that the images in this article are still well recognizable under Color Blindness.

Fig1:

[Figure]

Fig2:

| Red-Weak/Protanomaly | Red-Blind/Protanopia | Monochromacy/Achromatopsia |
|---|---|---|
|
[Figure]
 | | |
| Green-Weak/Deuteranomaly | Green-Blind/Deuteranopia | Blue Cone Monochromacy |
| | | |
| Blue-Weak/Tritanomaly | Blue-Blind/Tritanopia | |
| | | |

Fig3:

| Red-Weak/Protanomaly | Red-Blind/Protanopia | Monochromacy/Achromatopsia |
|---|---|---|
|
[Figure]
 | | |
| Green-Weak/Deuteranomaly | Green-Blind/Deuteranopia | Blue Cone Monochromacy |
| | | |
| Blue-Weak/Tritanomaly | Blue-Blind/Tritanopia | |
| | | |

Fig4:

[Figure]

| Red-Weak/Protanomaly | Red-Blind/Protanopia | Monochromacy/Achromatopsia |
| --- | --- | --- |
| Green-Weak/Deuteranomaly | Green-Blind/Deuteranopia | Blue Cone Monochromacy |
| Blue-Weak/Tritanomaly | Blue-Blind/Tritanopia | |

Fig5:

| Red-Weak/Protanomaly | Red-Blind/Protanopia | Monochromacy/Achromatopsia |
|---|---|---|
| |
[Figure]
 | |
| Green-Weak/Deuteranomaly | Green-Blind/Deuteranopia | Blue Cone Monochromacy |
| Blue-Weak/Tritanomaly | Blue-Blind/Tritanopia | |

Fig6:

| Red-Weak/Protanomaly | Red-Blind/Protanopia | Monochromacy/Achromatopsia |
|---|---|---|
|
[Figure]
 | | |

| Green-Weak/Deuteranomaly | Green-Blind/Deuteranopia | Blue Cone Monochromacy |
|---|---|---|

| Blue-Weak/Tritanomaly | Blue-Blind/Tritanopia | |
|---|---|---|

Fig7:

| Red-Weak/Protanomaly | Red-Blind/Protanopia | Monochromacy/Achromatopsia |
|---|---|---|

| Green-Weak/Deuteranomaly | Green-Blind/Deuteranopia | Blue Cone Monochromacy |
|---|---|---|

| Blue-Weak/Tritanomaly | Blue-Blind/Tritanopia | |
|---|---|---|

Fig8:

| Red-Weak/Protanomaly | Red-Blind/Protanopia | Monochromacy/Achromatopsia |
|---|---|---|
| | | |
| Green-Weak/Deuteranomaly | Green-Blind/Deuteranopia | Blue Cone Monochromacy |
| | | |
| Blue-Weak/Tritanomaly | Blue-Blind/Tritanopia | |

[Figure]

Fig9:

[Figure]

Fig10:

[Figure]